# Generalization Bounds for Out-of-distribution Generalization

**Xin Zou** [1]  **Xiuwen Gong** [1]  **Weiwei Liu** [2]

## Abstract

Out-of-distribution (OOD) generalization has attracted increasing research attention in recent years, owing to its promising empirical results in real-world applications. However, theoretical studies on OOD generalization remain limited, particularly with respect to lower bounds on the generalization error. To better understand how source data contributes to improved OOD generalization performance, we take an initial step toward establishing a lower bound on the OOD generalization error, and subsequently investigate upper bounds from the perspective of statistical learning theory. Interestingly, we find that under the RCS and empirical-RCS conditions, simply minimizing the average empirical risk over the source domains can yield a nearly optimal error rate (up to a logarithmic factor) *without* requiring knowledge or estimation of distributional parameters or the discrepancy between source and target domains. This conditional result provides a theoretical perspective on the surprising phenomenon observed in DomainBed (Gulrajani & Lopez-Paz, 2021), where carefully designed OOD generalization algorithms fail to outperform the simple empirical risk minimization (ERM) algorithm. Our results also imply a no-free-lunch theorem and provide an optimistic bound for OOD generalization.

## 1. Introduction

### 1.1. Background

Recent years have witnessed the remarkable success of modern machine learning techniques in many applications (Ma et al., 2026; He et al., 2016; Ma et al., 2025; Vaswani et al., 2017; Ma et al., 2024). A fundamental assumption in machine learning is that the training and test data are drawn from the same underlying distribution. However, this assumption is often violated in many practical applications. The test distribution is influenced by a range of factors, such as the distributional shifts across the photos caused by different cameras in image classification tasks, variations in speakers' voices in voice recognition tasks, and scene variations in self-driving tasks (Nagarajan et al., 2021). Therefore, there is now a rapidly growing body of research with a focus on generalizing to unseen distributions, namely, **out-of-distribution (OOD)** generalization (Shen et al., 2021). However, despite the significant effort in the last decade, theoretical studies on OOD generalization remain limited, especially lower bounds for the generalization error on the target distribution. Moreover, current theoretical works (Arjovsky et al., 2019; Rosenfeld et al., 2021; Zhou et al., 2022; Chen et al., 2022; Wang et al., 2022; Garg et al., 2021; Li et al., 2022; Blanchard et al., 2011; 2021; Ye et al., 2021; Nagarajan et al., 2021; Lin et al., 2024; Zou et al., 2024; Tang et al., 2025) focus on deriving upper bounds for the target risk based on certain assumptions about the target distribution and then designing algorithms to minimize these bounds. None of them explore the upper and lower bounds for the target excess risk in the same spirit as the statistical learning theory community (Vapnik & Chervonenkis, 1974; 2015; Shalev-Shwartz & Ben-David, 2014). Motivated by this gap, we study the following fundamental questions for OOD generalization:

> **Fundamental Questions**
> 1. What is (the upper and lower bounds of) the rate of the excess risk on the target domain?
> 2. How to design an algorithm that achieves (nearly) optimal rate of target excess risk?

In this paper, we answer the above questions by investigating the lower bounds for OOD generalization and exploring upper bounds (through algorithm design). We also show some implications of our results.

### 1.2. Overview of the Main Results

We consider binary classification where $\mathcal{X}$ is the input space and $\mathcal{Y} = \{-1, +1\}$ is the label space. We let $d$ be the Vapnik-Chervonenkis (VC) dimension (Vapnik &

---

[1]School of Computer Science, Wuhan University, China [2]Institute of Big Data, Fudan University, China. Correspondence to: Xiuwen Gong <gongxiuwen@gmail.com>, Weiwei Liu <liuweiwei863@gmail.com>.

*Proceedings of the 43$^{rd}$ International Conference on Machine Learning*, Seoul, South Korea. PMLR 306, 2026. Copyright 2026 by the author(s).

Chervonenkis, 2015) (Definition 3.5) of a hypothesis class $\mathcal{H} \subseteq \mathcal{Y}^{\mathcal{X}}$. In OOD generalization, we have access to $K$ source distributions $P_1, \ldots, P_K$, and wish to design an algorithm that achieves good performance on the target distribution $Q$, where the performance of a hypothesis $h \in \mathcal{H}$ is measured by the excess risk $\mathcal{E}_Q(h)$ (Section 3) under the 0-1 loss with respect to $\mathcal{H}$.

For $i \in \{1, \ldots, K\}$, we use the Bernstein Class Condition (BCC) parameter (Definition 3.1) $\beta_i$ to characterize the level of noise in the source domain $P_i$, and use the transfer-exponent (Definition 3.4) $\rho_i$ to describe the ability to transfer information from the $i$-th source domain $P_i$ to the target domain $Q$. Define $h_{P_i}^* = \underset{h \in \mathcal{H}}{\arg\min} \, \mathcal{E}_{P_i}(h)$ for each source domain $P_i$ and $h_Q^* = \underset{h \in \mathcal{H}}{\arg\min} \, \mathcal{E}_Q(h)$ for the target domain $Q$. Let $\alpha_i = \mathcal{E}_Q(h_{P_i}^*)$ for $i \in [K]$. We assume having access to $n_i$ training examples from $P_i$. Let $S = \{(x_{ij}, y_{ij})\}_{i \in [K], j \in [n_i]}$ be the whole training set, where $(x_{ij}, y_{ij})$ is the $j$-th example from $P_i$. We now present our main results informally; the formal results are shown in Sections 4 and 5.

**Theorem 1.1** (Lower Bound, Informal Version of Theorem 4.2)**.** *There is some constant $c > 0$, for any algorithm $\mathcal{A}$, there exists a distribution tuple $(P_1, \ldots, P_K, Q)$ with distributional parameters $(\beta_1, \ldots, \beta_K; \rho_1, \ldots, \rho_K; \alpha_1, \ldots, \alpha_K)$ such that with probability at least $\frac{3-2\sqrt{2}}{8}$ over the choice of the training set $S$,*

$$\mathcal{E}_Q(\mathcal{A}(S)) \geq c \cdot \min_{i \in [K]} \left\{ \left( \frac{d}{n_i} \right)^{\frac{1}{(2-\beta_i)\rho_i}} + \alpha_i \right\}.$$

Theorem 1.1 shows that the worst-case (with respect to the choice of distribution tuples with distributional parameters $(\beta_1, \ldots, \beta_K; \rho_1, \ldots, \rho_K; \alpha_1, \ldots, \alpha_K)$) target excess risk is of order $\Omega \left( \min_{i \in [K]} \left\{ \left( \frac{d}{n_i} \right)^{\frac{1}{(2-\beta_i)\rho_i}} + \alpha_i \right\} \right)$. In Lemma 5.1, we show that if we run ERM on the $i$-th source domain, the worst-case target excess risk is of order $\tilde{O} \left( \left( \frac{d}{n_i} \right)^{\frac{1}{(2-\beta_i)\rho_i}} + \alpha_i \right)$, which suggests that if we can find $i^*$ that reaches the minimum in the lower bound, we can then obtain a nearly optimal algorithm by running ERM on the $i^*$-th source domain. Unfortunately, choosing $i^*$ requires knowledge of the distributional parameters $(\beta_1, \ldots, \beta_K; \rho_1, \ldots, \rho_K; \alpha_1, \ldots, \alpha_K)$ which is not available in practice. Under the RCS and empirical-RCS conditions introduced in Section 5, we can design a nearly optimal algorithm without any information about the distributional parameters.

**Theorem 1.2** (Upper Bound, Informal Version of Theorem 5.12)**.** *Under the RCS and empirical-RCS conditions in Section 5, where $\alpha_1 = \cdots = \alpha_K = 0$, there exists an algorithm $\mathcal{A}_{ERM}$, for any distributional parame-*

*ters $(\beta_1, \ldots, \beta_K; \rho_1, \ldots, \rho_K)$ and any distribution tuple $(P_1, \ldots, P_K, Q)$ with such parameters, with probability at least $1 - \delta$ over the choice of the training set $S$, we have*

$$\mathcal{E}_Q(\mathcal{A}_{ERM}(S)) \leq \tilde{O} \left( \min_{i \in [K]} \left\{ \left( \frac{d}{n_i} \right)^{\frac{1}{(2-\beta_i)\rho_i}} \right\} \right).$$

Theorems 1.1 and 1.2 show that under the conditions in Theorem 1.2, the worst-case target excess risk is of order $\tilde{\Theta} \left( \min_{i \in [K]} \left\{ \left( \frac{d}{n_i} \right)^{\frac{1}{(2-\beta_i)\rho_i}} \right\} \right)$, and algorithm $\mathcal{A}_{ERM}$ is nearly optimal within this RCS / empirical-RCS regime. Interestingly, the algorithm $\mathcal{A}_{ERM}$ is very simple and merely minimizes the average source empirical risk (for more information, please refer to Algorithm 5).

Our **contributions** can be summarized as follows:

- A lower bound for the excess risk of all possible algorithms on the target distribution is presented through a minimax argument (Theorem 4.2).

- We explore the upper bounds for OOD generalization by analyzing the error bounds of the learning algorithms (Section 5).

- We show some implications of our upper and lower bounds, which help us better understand OOD generalization. (1) A no-free-lunch theorem for OOD generalization is presented in Remark 4.4: our lower bound shows that we cannot expect a single algorithm to do well on all OOD generalization tasks. (2) Our upper and lower bounds show that, in the RCS / empirical-RCS regime, the ERM algorithm is nearly optimal in terms of the excess risk rate (Theorem 5.12, Corollary 5.14). (3) The near optimality of ERM explains the surprising phenomenon found in DomainBed (Gulrajani & Lopez-Paz, 2021) (Remark 5.15): the carefully designed OOD generalization algorithms cannot outperform the simple ERM algorithm.

The remainder of this article is structured as follows: § 2 presents related works. § 3 introduces notations, setups, and some definitions. § 4 presents the lower bounds and § 5 presents the upper bounds. The conclusions are presented in the last section.

**Conflict of Interest Disclosure.** The authors declare no financial conflicts of interest or other substantive conflicts of interest that could reasonably be perceived to influence this work.

## 2. Related Works

OOD generalization has emerged as a critical challenge in machine learning, particularly as models are increasingly

deployed in environments where training and test distributions differ. In this section, we review key contributions in the literature, focusing on theoretical frameworks and error bounds for OOD generalization.

## 2.1. OOD Generalization

OOD generalization aims to train a model with data from the training distributions (e.g., domains, in this paper, we use the terminology "distribution" and "domain" interchangeably) so that it is capable of generalizing to an unseen target distribution. A large number of algorithms have been developed that aim to improve OOD generalization. A line of work focuses on minimizing the discrepancies between the training distributions (Li et al., 2018b; Ganin et al., 2016; Sun & Saenko, 2016; Albuquerque et al., 2021). Meta-learning domain generalization (Li et al., 2018a, MLDG) leverages the meta-learning approach and simulates train/test distributional shift during training by synthesizing virtual testing domains within each mini-batch. Sagawa et al. (2020, GroupDRO) studies applying distributionally robust optimization (DRO) (Gao & Kleywegt, 2023) to learn models that instead minimize the worst-case training loss over a set of pre-defined groups. However, according to Gulrajani & Lopez-Paz (2021), many OOD generalization algorithms cannot do better than ERM, indicating the emergency to theoretically study generalization behaviors of OOD generalization methods.

From the theoretical perspective, the seminal work Arjovsky et al. (2019) studies the behavior of invariant risk minimization (IRM) in OOD generalization and shows generalization theory for IRM in terms of the structural equation models (SEM). Following this line, there are some works discussing the IRM algorithm Rosenfeld et al. (2021); Zhou et al. (2022) and the environment complexity Chen et al. (2022); Wang et al. (2022). In addition to the IRM-based line, there are some other perspectives for OOD generalization: Garg et al. (2021); Li et al. (2022) consider the case that both the training and test domains are drawn from some distribution over domains. Blanchard et al. (2011; 2021) view OOD generalization as a kind of supervised learning problem by augmenting the original feature space with the marginal distribution of feature vectors. Eastwood et al. (2022) consider generalization bounds for the quantile risk minimization (QRM) algorithm. Zou & Liu (2023) study the adversarial robustness of OOD generalization models. Ye et al. (2021) formalize OOD generalization problem by quantifying the feature distribution. Zou & Liu (2024) propose a conformal prediction method for OOD generalization. Nagarajan et al. (2021) study how spurious features lead to the failure of OOD generalization, while Lin et al. (2024) show that ensemble methods can attain superior OOD performance by utilizing a diverse set of spurious features. The above works mainly focus on deriving upper bounds of the target risk, however, this paper investigates both upper and lower bounds in OOD generalization, and proposes a nearly optimal algorithm for OOD generalization.

## 2.2. Transfer Exponent

The transfer-exponent is firstly proposed by (Hanneke & Kpotufe, 2019) as a measure of the extent to which information from the source domain can be transferred to the target domain. Hanneke & Kpotufe (2019) provide generalization bounds for transfer learning with the transfer-exponent, and the following work (Hanneke et al., 2023) studies the limits of model selection under transfer learning in terms of the transfer exponent. Compared to transfer learning, OOD generalization has no access to the target data and have multiple source domains. Thus it is unclear whether the transfer-exponent can be used to analyze OOD generalization. This work provides an affirmative answer. In Section C, we argue that our results are not simple generalizations of those in (Hanneke & Kpotufe, 2019; Hanneke et al., 2023) and discuss the novelty of our results. Examples of calculating the transfer exponent and the difference between the transfer exponent and other discrepancies of distributions can be found in (Hanneke & Kpotufe, 2019).

## 3. Preliminaries

In this section, we introduce the key concepts and definitions necessary for understanding OOD generalization.

**Notations.** We use $[n]$ to denote $\{1, \ldots, n\}$ for some integer $n \in \mathbb{N}$. We denote scalars and vectors with lowercase letters and lowercase bold letters respectively. We use $\mathbb{1}[\cdot]$ as the indicator function where $\mathbb{1}[A] = 1$ when $A$ is true, otherwise $\mathbb{1}[A] = 0$. For two vectors $\mathbf{x}, \mathbf{y} \in \mathbb{R}^n$, we define the Hamming distance between $\mathbf{x}, \mathbf{y}$ as $H(\mathbf{x}, \mathbf{y}) = \sum_{i=1}^{n} \mathbb{1}[x_i \neq y_i]$, where $x_i, y_i$ are the $i$-th element of $\mathbf{x}$ and $\mathbf{y}$, respectively.

**Setups.** Let $\mathcal{X}$ be the input space and $\mathcal{Y}$ be the label space. In this paper, we consider $\mathcal{Y} = \{-1, +1\}$. We use $\ell : \mathcal{Y} \times \mathcal{Y} \to \mathbb{R}_+$ as the loss function. In this paper, we use the 0-1 loss, i.e., $\ell(y, y') \coloneqq \mathbb{1}[y \neq y']$. We consider learning with the hypothesis class $\mathcal{H} \subseteq \mathcal{Y}^{\mathcal{X}}$. Given probability measure $D$ on $\mathcal{X} \times \mathcal{Y}$, the generalization error of hypothesis $h \in \mathcal{H}$ with respect to $\ell$ is defined as $R_D(h) \coloneqq \mathbb{E}_{(x,y) \sim D}[\ell(h(x), y)]$, the empirical error is defined similarly with respect to the empirical distribution $\hat{D}$. The excess risk of $h$ is defined as $\mathcal{E}_D(h) \coloneqq R_D(h) - \inf_{h' \in \mathcal{H}} R_D(h')$. For simplicity, we define $\mathcal{E}_D(h, h') \coloneqq \mathcal{E}_D(h) - \mathcal{E}_D(h')$. We let $D_X, D_{Y|X}$ denote the marginal and conditional distributions of $D$. OOD generalization assumes (Li et al., 2018b; Ganin et al., 2016; Li et al., 2018c; Sun & Saenko, 2016; Albuquerque et al., 2021; Li et al., 2018a; Gulrajani & Lopez-Paz, 2021; Arjovsky et al., 2019; Rosenfeld et al., 2021; Zhou et al., 2022; Garg et al., 2021; Li et al., 2022; Eastwood et al., 2022)

that we can obtain datasets from $K$ source distributions $P_1, \ldots, P_K$, our goal is to learn a hypothesis that generalizes well on an unknown target distribution $Q$, i.e., to achieve low excess risk on $Q$.

We adopt a classical noise condition to characterize the level of noise in the data.

**Definition 3.1** (The Bernstein Class Condition, Bartlett & Mendelson, 2006; Massart & Nédélec, 2006; Tsybakov, 2004; Mammen & Tsybakov, 1999; Hanneke & Kpotufe, 2019; Hanneke et al., 2023)**.** Let $h_D^* := \arg\min_{h \in \mathcal{H}} R_D(h)$. We say that the probability measure $D$ satisfies the *Bernstein Class Condition (BCC)* with parameters $(\beta, c)$, where $\beta \in [0, 1]$ and $c > 0$, if for all $h \in \mathcal{H}$, we have:

$$D_X(h \neq h_D^*) \leq c \cdot \mathcal{E}_D^\beta(h). \tag{1}$$

The condition in Definition 3.1 assumes that there exists one and only one minimizer of the risk $R_D(h)$. We can relax this assumption by allowing multiple minimizers. We define the relaxed version of the Bernstein Class Condition as follows.

**Definition 3.2** (The Relaxed Bernstein Class Condition)**.** Let $\mathcal{H}_D := \arg\min_{h \in \mathcal{H}} R_D(h) \subseteq \mathcal{H}$ be the set of minimizers of $R_D$. We say that the probability measure $D$ satisfies the *Relaxed Bernstein Class Condition (RBCC)* with parameters $(\beta, c)$, where $\beta \in [0, 1]$ and $c > 0$, if for all $h \in \mathcal{H}$, we have:

$$\inf_{h^* \in \mathcal{H}_D} D_X(h \neq h^*) \leq c \cdot \mathcal{E}_D^\beta(h). \tag{2}$$

*Remark* 3.3. Note that the above definition is a generalization of the traditional dichotomy of the learning rate for the realizable (with the rate $O\left(n^{-1}\right)$) and agnostic (with the rate $O\left(n^{-1/2}\right)$) cases in the probably approximately correct (PAC) learning framework (Valiant, 1984). It is easy to verify that a realizable distribution satisfies the RBCC with parameters $(1, 1)$, while an agnostic distribution satisfies the RBCC with parameters $(0, 1)$ if we slightly abuse the notation and allow $0 \leq 0^0$. Rather than the dichotomy of $O\left(\frac{1}{n}\right)$ and $O\left(\frac{1}{\sqrt{n}}\right)$, Definition 3.2 allows us to describe an excess risk rate of order $O\left(\left(\frac{1}{n}\right)^{\frac{1}{2-\beta}}\right)$ if we set $P = Q$ in Lemma 5.1.

Many works (Li et al., 2018b; Ganin et al., 2016; Li et al., 2018c; Sun & Saenko, 2016) on OOD generalization use discrepancies between distributions to measure the difference between the training distributions and try to minimize the discrepancies between source domains in order to improve the generalization performance. For example, Li et al. (2018b) uses the Maximum Mean Discrepancy (MMD) to measure the differences; Ganin et al. (2016); Li et al. (2018c) apply the $\mathcal{H}$-divergence as a measure; Sun & Saenko (2016)

defines CORAL, the distance between the second-order statistics (covariances) of the features from different domains, to measure the differences between domains. On the contrary, in this paper, we use the transfer exponent to describe the ability to transfer information from one distribution to another while not optimizing the transfer exponent.

**Definition 3.4** (Transfer Exponent, Hanneke & Kpotufe, 2019; Hanneke et al., 2023)**.** Let $P, Q$ be probability measures on $\mathcal{X} \times \mathcal{Y}$ and $\mathcal{H}$ be a hypothesis class. We call $0 < \rho \leq \infty$ a *transfer exponent* from $P$ to $Q$ with respect to $\mathcal{H}$ if there exists $C_\rho > 0$ such that for all $h \in \mathcal{H}$, we have:

$$\mathcal{E}_Q\left(h, h_P^*\right) \leq C_\rho \cdot \mathcal{E}_P^{\frac{1}{\rho}}(h, h_P^*). \tag{3}$$

We say that $\rho$ is minimal when no $0 < \rho' < \rho$ is a transfer exponent from $P$ to $Q$ with respect to $\mathcal{H}$.

In addition to the Bernstein Class Condition and the transfer exponent, we introduce a parameter to characterize the complexity of a hypothesis class. Here we use the well-known VC dimension (Vapnik & Chervonenkis, 2015) from statistical learning theory.

**Definition 3.5** (VC Dimension, Vapnik & Chervonenkis, 2015)**.** Let $\mathcal{H}$ be a hypothesis class, a sequence $\{x_1, \cdots, x_n\}$ of points in $\mathcal{X}$ is said to be shattered by $\mathcal{H}$ if $\forall y_1, \ldots, y_n \in \mathcal{Y}, \exists h \in \mathcal{H}$ such that $\forall i \in [n], h(x_i) = y_i$. The *VC dimension* of $\mathcal{H}$ is defined as the cardinality of the largest set that can be shattered by $\mathcal{H}$; if no such largest cardinality exists, the VC dimension is said to be infinite. We denote the VC dimension of $\mathcal{H}$ as $\text{VCdim}(\mathcal{H})$.

## 4. Lower Bounds

**Definition 4.1** ($\mathcal{P}$ Class)**.** Let $\rho_1, \ldots, \rho_K \in (0, \infty], \beta_1, \ldots, \beta_K, \beta \in [0, 1], \alpha_1, \ldots, \alpha_K \in [0, 1], C > 0$. For a distribution tuple $(P_1, \ldots, P_K, Q)$, we say that $(P_1, \ldots, P_K, Q) \in \mathcal{P}(\mathcal{H}; \rho_1, \ldots, \rho_K; \beta_1, \ldots, \beta_K; \beta; \alpha_1, \ldots, \alpha_K; C)$ if the following conditions hold:

- For all $i \in [K]$, $P_i$ satisfies the RBCC with parameters $(\beta_i, c_i)$ with $c_i \leq C$. $Q$ satisfies the RBCC with parameters $(\beta, c)$ with $c \leq C$.

- For all $i \in [K]$, the transfer exponent from $P_i$ to $Q$ with respect to $\mathcal{H}$ is $\rho_i$, with the corresponding constant $C_{\rho_i} \leq C$.

- For all $i \in [K]$, $\mathcal{E}_Q\left(h_{P_i^*}\right) \leq \alpha_i$, where $h_{P_i^*} \in \mathcal{H}_{P_i}$ is the minimizer of $R_{P_i}(\cdot)$.

When the context is clear, we denote $\mathcal{P}(\mathcal{H}; \rho_1, \ldots, \rho_K; \beta_1, \ldots, \beta_K; \beta; \alpha_1, \ldots, \alpha_K; C)$ as $\mathcal{P}$ for simplicity.

**Theorem 4.2** (Lower Bound). *Let $\mathcal{H}$ be a hypothesis class with VC dimension $VCdim(\mathcal{H}) = d \geq 9$. Fix any $\rho_1, \ldots, \rho_K \in [1, \infty], \beta_1, \ldots, \beta_K, \beta \in [0, 1], \alpha_1, \ldots, \alpha_K \in [0, 1]$. Let $\hat{h} = \hat{h}(S_{P_1}, \ldots, S_{P_K})$ denote any (possibly improper) classifier with access to $K$ independent labeled samples $S_{P_1} \sim P_1^{n_1}, \ldots, S_{P_K} \sim P_K^{n_K}$. Suppose that one of $\{n_1, \ldots, n_K\}$ is sufficiently large so that*

$$\varepsilon(n_1, \ldots, n_K) := \min_{1 \leq i \leq K} \left\{ \left( \frac{d}{n_i} \right)^{\frac{1}{(2-\beta_i)\rho_i}} \right\} \leq 1.$$

*Then, for any $\hat{h}$, there exists $(P_1, \ldots, P_K, Q) \in \mathcal{P}(\mathcal{H}; \rho_1, \ldots, \rho_K; \beta_1, \ldots, \beta_K; \beta; \alpha_1, \ldots, \alpha_K; 1)$ and a universal constant $c > 0$ such that*

$$\mathbb{P}_{S_{P_1} \sim P_1^{n_1}, \ldots, S_{P_K} \sim P_K^{n_K}} \left( \mathcal{E}_Q\left(\hat{h}\right) > c \cdot s_K \right) \geq \frac{3 - 2\sqrt{2}}{8},$$

*where*

$$s_K := \min_{1 \leq i \leq K} \left\{ \left( \frac{d}{n_i} \right)^{\frac{1}{(2-\beta_i)\rho_i}} + \alpha_i \right\}.$$

*Remark* 4.3 (Reduction to PAC Learning Results). According to Theorem 4.2, we can conclude that if the VC dimension of the hypothesis class $\mathcal{H}$ has infinite VC dimension, then the target excess risk remains bounded away from zero, even when $\alpha_1 = \cdots = \alpha_K = 0$. The above results implies that a finite VC dimension is the necessary condition to have a vanishing (tends to zero as sample size $n \rightarrow \infty$) excess risk on the target distribution. By choosing $K = 1$ and $P_1 = Q$, we have $\rho_1 = 1$ and $\mathcal{E}_Q\left(h_{P_1^*}\right) = 0$. In this case, we recover a lower bound for traditional supervised learning of order $\Omega\left( \left(\frac{d}{n_1}\right)^{\frac{1}{2-\beta_1}} \right)$.

*Remark* 4.4 (A No-Free-Lunch Theorem for OOD Generalization). The lower bound in Theorem 4.2 presents an impossibility result in the absence of any assumptions on the target distribution. By setting $\rho_1 = \cdots = \rho_K \rightarrow \infty$, the generalization error remains bounded away from zero regardless of the amount of training data or the choice of algorithm—even when there exists a hypothesis $h^* \in \mathcal{H}$ that simultaneously minimizes both the source and target risks (i.e., $\alpha_1 = \cdots = \alpha_K = 0$). So the above argument holds even when Assumption 5.5 holds. Since the above result shares a similar spirit with the no-free-lunch theorem (Shalev-Shwartz & Ben-David, 2014) in statistical learning theory, we refer to it as the no-free-lunch theorem for OOD generalization, which tells us that we cannot hope a single algorithm to solve all the OOD generalization problems and we cannot expect an algorithm to generalize well on all possible target distributions. Algorithms must be designed according to specific application scenarios.

*Proof Sketch of Theorem 4.2.* We provide a proof sketch of Theorem 4.2 here. The full proof can be found in Section A. Our lower bound relies on Lemma A.3, an extension of Fano's inequality.

For simplicity, we define $\varepsilon_i := \left( \frac{d}{n_i} \right)^{\frac{1}{(2-\beta_i)\rho_i}}$, then $s_K = \min \{\varepsilon_1 + \alpha_1, \ldots, \varepsilon_K + \alpha_K\}$. We define $r_j = \max\{\varepsilon_j, \alpha_j\}$ for $j \in [K]$ and define $s'_K = \min\{r_1, \ldots, r_K\}$. Then, since $s'_K \leq s_K \leq 2s'_K$, we only need to prove the lower bound is of order $\Omega\left(s'_K\right)$. We define $s = c_1 \cdot s'_K \in [0, 1]$ where $c_1 \in (0, 1)$ is a constant to be determined later. So it suffices to prove that the lower bound is of order $\Omega(s)$.

Our construction is based on a set $\mathcal{S} = \{x_0, x_1, \ldots, x_{d-1}\} \subseteq \mathcal{X}$ that is shattered by $\mathcal{H}$. We construct distributions supported on $\mathcal{S}$, then we only need to consider $\tilde{\mathcal{H}}$, the projection of $\mathcal{H}$ onto $\mathcal{S}$. To apply Lemma A.3, we define the Hamming distance $H(\cdot, \cdot)$ on $\tilde{\mathcal{H}}$, where

$$H(h, h') = \sum_{i=1}^{d-1} \mathbb{1}[h(x_i) \neq h'(x_i)], \quad \forall h, h' \in \tilde{\mathcal{H}}.$$

It is easy to verify that $H(\cdot, \cdot)$ is a semi-metric. Let $\boldsymbol{\sigma} \in \{-1, +1\}^{d-1}$, we will construct a family of distribution tuples $(P_{1,\boldsymbol{\sigma}}, \cdots, P_{K,\boldsymbol{\sigma}}, Q_{\boldsymbol{\sigma}})$ indexed by $\boldsymbol{\sigma} \in \{-1, +1\}^{d-1}$ and then apply Lemma A.3 to prove the lower bound. For simplicity, we use $h_{\boldsymbol{\sigma}}$ to present the hypothesis in $\tilde{\mathcal{H}}$ that where $h_{\boldsymbol{\sigma}}(x_0) = 1$ and $h_{\boldsymbol{\sigma}}(x_i) = \boldsymbol{\sigma}_i$ for $i \in [d-1]$; and use $h'_{\boldsymbol{\sigma}}$ to present the hypothesis in $\tilde{\mathcal{H}}$ that where $h'_{\boldsymbol{\sigma}}(x_0) = -1$ and $h'_{\boldsymbol{\sigma}}(x_i) = \boldsymbol{\sigma}_i$ for $i \in [d-1]$. Let $\boldsymbol{\eta}_1, \ldots, \boldsymbol{\eta}_K \in \{-1, +1\}^{d-1}$ be $K$ vectors. Our construction of the distribution tuples is as follows:

**Construction of the Distribution $Q_{\boldsymbol{\sigma}}$.** Let $Q_{\boldsymbol{\sigma}} = Q_X \times Q_{\boldsymbol{\sigma}, Y|X}$. We set $Q_X(x_0) = 1 - s^\beta$ and $Q_X(x_i) = \frac{s^\beta}{d-1}$ for $i \in [d-1]$. For $Q_{\boldsymbol{\sigma}, Y|X}$, we set $Q_{\boldsymbol{\sigma}, Y|X=x_0}(Y=1) = \frac{1}{2}$ and $Q_{\boldsymbol{\sigma}, Y|X=x_i}(Y=1) = \frac{1}{2} + \frac{\boldsymbol{\sigma}_i}{2} \cdot s^{1-\beta}$ for $i \in [d-1]$. In this case, $\{h_{\boldsymbol{\sigma}}, h'_{\boldsymbol{\sigma}}\} \in \arg\min_{h \in \tilde{\mathcal{H}}} R_{Q_{\boldsymbol{\sigma}}}(h)$.

**Construction of the Distribution $P_{j,\boldsymbol{\sigma}}$ for $j \in [K]$.** The construction of $P_{j,\boldsymbol{\sigma}}$ depends on the exact value of $r_j$. We consider the following two cases:

- If $r_j = \varepsilon_j$, we set $P_{j,\boldsymbol{\sigma}} = P_{j,X} \times P_{j,\boldsymbol{\sigma}, Y|X}$. We set $P_{j,X}(x_0) = 1 - s^{\rho_j \beta_j}$ and $P_{j,X}(x_i) = \frac{s^{\rho_j \beta_j}}{d-1}$ for $i \in [d-1]$. For $P_{j,\boldsymbol{\sigma}, Y|X}$, we set $P_{j,\boldsymbol{\sigma}, Y|X=x_0}(Y=1) = \frac{1}{2}$ and $P_{j,\boldsymbol{\sigma}, Y|X=x_i}(Y=1) = \frac{1}{2} + \frac{\boldsymbol{\sigma}_i}{2} \cdot s^{\rho_j(1-\beta_j)}$ for $i \in [d-1]$. In this case, $\{h_{\boldsymbol{\sigma}}, h'_{\boldsymbol{\sigma}}\} \in \arg\min_{h \in \tilde{\mathcal{H}}} R_{P_{j,\boldsymbol{\sigma}}}(h)$.

- If $r_j = \alpha_j$, we set $P_{j,\boldsymbol{\sigma}} = P'_j = P'_{j,X} \times P'_{j,Y|X}$. We set $P'_{j,X}(x_0) = 0$ and $P'_{j,X}(x_i) = \frac{1}{d-1}$ for $i \in [d-1]$.

For $P'_{j,Y|X}$, we set $P_{j,Y|X=x_i}(Y = \boldsymbol{\eta}_{j,i}) = 1$. In this case, $\{h_{\boldsymbol{\eta}_j}, h'_{\boldsymbol{\eta}_j}\} \in \arg\min\limits_{h \in \tilde{\mathcal{H}}} R_{P'_j}(h)$.

With some efforts, we can prove that the distribution tuples $(P_{1,\boldsymbol{\sigma}}, \cdots, P_{K,\boldsymbol{\sigma}}, Q_{\boldsymbol{\sigma}})$ satisfy the conditions in $\mathcal{P}(\mathcal{H}; \rho_1, \ldots, \rho_K; \beta_1, \ldots, \beta_K; \beta; \alpha_1, \ldots, \alpha_K; 1)$. Then, by Lemma A.4, there exists a subset $\Sigma \subseteq \{-1, +1\}^{d-1}$ with $M := |\Sigma| \geq 2^{\frac{d-1}{8}}$ such that $H(\boldsymbol{\sigma}_i, \boldsymbol{\sigma}_j) \geq \frac{d-1}{8}$ for any $\boldsymbol{\sigma}_i, \boldsymbol{\sigma}_j \in \Sigma$. It shows that for any $\boldsymbol{\sigma}, \boldsymbol{\sigma}' \in \Sigma$, we have: $\mathcal{E}_{Q_{\boldsymbol{\sigma}}}(h_{\boldsymbol{\sigma}'}) = \frac{H(\boldsymbol{\sigma}, \boldsymbol{\sigma}')}{d-1} \cdot s \geq \frac{s}{8}$, which is exactly of order $\Omega(s)$. Let $\Pi_{\boldsymbol{\sigma}} = P_{1,\boldsymbol{\sigma}}^{n_1} \times \cdots \times P_{K,\boldsymbol{\sigma}}^{n_K}$, then we can verify that for some proper choice of the constant $c_1$, $D_{\mathrm{KL}}(\Pi_{\boldsymbol{\sigma}} \| \Pi_{\boldsymbol{\sigma}'}) \leq \frac{1}{8}M$, then we can apply Lemma A.3 to prove the lower bound. $\square$

## 5. Upper Bounds

In this section, we explore upper bounds on the excess risk of OOD generalization. All the omitted proofs are provided in Section B.

**Lemma 5.1** (Excess Risk of the Single-Domain ERM Algorithm). *Let $\mathcal{H}$ be a hypothesis class with $VCdim(\mathcal{H}) = d$. Let $P$ be a probability measure on $\mathcal{X} \times \mathcal{Y}$ that satisfies the RBCC with parameters $(\beta, c_\beta)$ and let $Q$ be a probability measure on $\mathcal{X} \times \mathcal{Y}$ such that the transfer-exponent from $P$ to $Q$ is $\rho$ with parameter $C_\rho$. Define $h_P^* \in \arg\min\limits_{h \in \mathcal{H}} R_P(h)$ as one of the optimal classifiers for $P$. Let $S \sim P^n$ be a labeled sample of size $n$. Let $\hat{h} = \hat{h}(S) = \arg\min\limits_{h \in \mathcal{H}} R_{\hat{P}}(h)$ be the output of the empirical risk minimization algorithm. Then there exists a constant $c_\rho > 0$ such that: with probability at least $1 - \delta$ over the choice of $S$,*

$$\mathcal{E}_Q\left(\hat{h}\right) \leq \mathcal{E}_Q\left(h_P^*\right) + c_\rho \cdot A_n^{\frac{1}{\rho(2-\beta)}},$$

*where $A_n = \frac{d}{n} \ln\left(\frac{\max\{n,d\}}{d}\right) + \frac{\ln 1/(3\delta)}{n}$.*

According to Theorem 4.2 and Lemma 5.1, we find that maybe the best we can do is to identify the source domain with smallest $\mathcal{E}_Q(h_P^*) + c_\rho \cdot A_n^{\frac{1}{\rho(2-\beta)}}$ and run the single-domain ERM (SD-ERM) algorithm on such domain, which can be achieved by the following oracle algorithm.

**Theorem 5.2** (Oracle Algorithm Achieves Nearly Optimal Rate). *Let $\mathcal{H}$ be a hypothesis class with $VCdim(\mathcal{H}) = d$. Let $(P_1, \ldots, P_K, Q) \in \mathcal{P}(\mathcal{H}; \rho_1, \ldots, \rho_K; \beta_1, \ldots, \beta_K; \beta; \alpha_1, \ldots, \alpha_K; C)$. Let $\hat{h}$ be the output of Algorithm 1. Then there exists a constant $C' > 0$ (depending on $\rho_i, \beta_i$ and $C$) such that, with probability at least $1 - \delta$,*

$$\mathcal{E}_Q\left(\hat{h}\right) \leq \min_{1 \leq i \leq K}\left\{\mathcal{E}_Q\left(h_{P_i}^*\right) + C' \cdot A_{n_i}^{\frac{1}{\rho_i(2-\beta_i)}}\right\} = \tilde{O}(s_K),$$

---

**Algorithm 1** The Oracle Algorithm

**Input:** $\mathcal{H}, \left\{S_{P_i}, \rho_i, \beta_i, c_{\rho,i}, \alpha_i = \mathcal{E}_Q\left(h_{P_i}^*\right)\right\}_{i=1}^K$.

Choose $i^* = \arg\min\limits_{i \in [K]}\left\{\alpha_i + c_{\rho,i} \cdot A_{n_i}^{\frac{1}{\rho_i(2-\beta_i)}}\right\}$.

Run ERM algorithm on $S_{P_{i^*}}$ to get $\hat{h}_{P_{i^*}} = \arg\min\limits_{h \in \mathcal{H}} R_{\hat{P}_{i^*}}(h)$.

**Output:** $\hat{h} = \hat{h}_{P_{i^*}}$.

---

**Algorithm 2** The AD-ERM Algorithm

**Input:** $\mathcal{H}, \{S_{P_i}\}_{i=1}^K$.

Arbitrarily choose $k \in [K]$.

Run ERM algorithm on $S_{P_k}$ to get $\hat{h}_{P_k} = \arg\min\limits_{h \in \mathcal{H}} R_{\hat{P}_k}(h)$.

**Output:** $\hat{h} = \hat{h}_{P_k}$.

---

*which implies that Algorithm 1 achieves the optimal rate up to a logarithm factor.*

Theorem 5.2 tells us that identifying a "good" source domain and running SD-ERM on the selected domain can be considered as a nearly optimal OOD generalization strategy. However, the oracle algorithm requires knowledge of the distributional parameters $\rho_i, \beta_i, c_{\rho,i}, \alpha_i$ to determine on which domain to run SD-ERM, which are typically unavailable in practice. In the absence of the distributional parameters, a simple alternative is to run SD-ERM on an arbitrary domain, which is called AD-ERM (arbitrary-domain ERM) in this paper.

**Theorem 5.3** (The Upper Bound for Algorithm 2). *Let $\mathcal{H}$ be a hypothesis class with $VCdim(\mathcal{H}) = d$. Let $(P_1, \ldots, P_K, Q) \in \mathcal{P}(\mathcal{H}; \rho_1, \ldots, \rho_K; \beta_1, \ldots, \beta_K; \beta; \alpha_1, \ldots, \alpha_K; C)$. Let $\hat{h}$ be the output of Algorithm 2. Then there exists a constant $C' > 0$ (depending on $\rho_i, \beta_i$ and $C$) such that, with probability at least $1 - \delta$,*

$$\mathcal{E}_Q\left(\hat{h}\right) \leq \max_{1 \leq i \leq K}\left\{\mathcal{E}_Q\left(h_{P_i}^*\right) + C' \cdot A_{n_i}^{\frac{1}{\rho_i(2-\beta_i)}}\right\}.$$

Note that the upper bound in Theorem 5.3 is suboptimal as it is of the order $\max\{\cdots\}$ rather than the $\min\{\cdots\}$ in the lower bound. This is because we do not know which source domain to select for running the SD-ERM algorithm. Algorithm 2 can be slightly improved by incorporating randomness. We define the randomized classifier based on $h_1, \ldots, h_m$ as follows: for all $i \in [m]$,

$$\mathrm{Rand}(h_1, \ldots, h_m)(x) = h_i(x) \text{ with probability } \frac{1}{m}.$$

**Algorithm 3** The Randomized ERM (R-ERM) Algorithm

**Input:** $\mathcal{H}, \{S_{P_i}\}_{i=1}^{K}$.
**for** $i = 1$ **to** $K$ **do**
    Set $\hat{h}_i = \underset{h \in \mathcal{H}}{\arg\min} R_{\hat{P}_i}(h)$.
**end for**
**Output:** $\hat{h}$.

---

**Algorithm 4** The Constrained ERM (C-ERM) Algorithm

**Input:** $\mathcal{H}, \{S_{P_i}\}_{i=1}^{K}$, constant $c$ in Lemma B.1.
**for** $j = 2$ **to** $K$ **do**
    Set

$$\mathcal{H}_j = \left\{ h \in \mathcal{H} : R_{\hat{P}_j}(h) - R_{\hat{P}_j}\left(h_{\hat{P}_j}\right) \right.$$
$$\left. \leq c\sqrt{\hat{P}_j\left(h \neq h_{\hat{P}_j}\right) A_{n_j, K}} + cA_{n_j, K} \right\}$$

**end for**
Select $\hat{h} \in \underset{h \in \cap_{j=2}^{K} \mathcal{H}_j}{\arg\min} R_{\hat{P}_1}(h)$

**Output:** $\hat{h}$.

---

**Theorem 5.4** (The Upper Bound for Algorithm 3). *Let $\mathcal{H}$ be a hypothesis class with $VCdim(\mathcal{H}) = d$. Let $(P_1, \ldots, P_K, Q) \in \mathcal{P}(\mathcal{H}; \rho_1, \ldots, \rho_K; \beta_1, \ldots, \beta_K; \beta; \alpha_1, \ldots, \alpha_K; C)$. Let $\hat{h}$ be the output of Algorithm 3. Then there exists a constant $C' > 0$ (depending on $\rho_i, \beta_i$ and $C$) such that, with probability at least $1 - \delta$,*

$$\mathcal{E}_Q\left(\hat{h}\right) \leq \frac{1}{K} \sum_{i=1}^{K} \left( \mathcal{E}_Q\left(h_{P_i}^*\right) + C' \cdot A_{n_i, K}^{\frac{1}{\rho_i(2-\beta_i)}} \right),$$

*where $A_{n,K} := \frac{d}{n} \ln\left(\frac{\max\{n, d\}}{d}\right) + \frac{\ln K/(3\delta)}{n}$.*

Unfortunately, randomness only improves the upper bound from $\max\{\cdots\}$ to $\text{avg}\{\cdots\}$, which is still not as tight as the $\min\{\cdots\}$ in the lower bound. In the following, we show that under mild assumptions, tighter upper bounds can be achieved.

**Assumption 5.5** (Generalization of the RCS Assumption from (Hanneke & Kpotufe, 2019)). Let $\mathcal{H}_{P_i} = \arg\min_{h \in \mathcal{H}} R_{P_i}(h)$ for $i \in [K]$ and $\mathcal{H}_Q = \arg\min_{h \in \mathcal{H}} R_Q(h)$. We assume that $\cap_{i=1}^{K} \mathcal{H}_{P_i} \cap \mathcal{H}_Q \neq \emptyset$, i.e., there exists some $h^* \in \mathcal{H}$ such that $h^* \in \arg\min_{h \in \mathcal{H}} R_{P_i}(h)$ for $i \in [K]$ and $h^* \in \arg\min_{h \in \mathcal{H}} R_Q(h)$.

*Remark 5.6.* As stated in (Hanneke & Kpotufe, 2019), Assumption 5.5 allows $P_{i,Y|X} \neq Q_{Y|X}$. However, it is not strictly weaker than the usually used Covariate-Shift assumption, which allows $h_{P_i}^* \neq h_Q^*$ when the Bayesian classifiers are not in $\mathcal{H}$. It readily follows from Assumption 5.5 that $\mathcal{E}_Q\left(h_{P_1}^*\right) = \cdots = \mathcal{E}_Q\left(h_{P_K}^*\right) = \mathcal{E}_Q\left(h_Q^*\right) = 0$.

**Lemma 5.7** (Transitivity of the Transfer-Exponent under Assumption 5.5). *Let $P, Q, R$ be distributions defined on $\mathcal{X} \times \mathcal{Y}$ and $\mathcal{H} \subseteq \mathcal{Y}^{\mathcal{X}}$ be a hypothesis class. Under Assumption 5.5, if the transfer-exponent from $P$ to $Q$ is $\rho_1$, with $C_{\rho_1} = C_1$ and the transfer-exponent from $Q$ to $R$ is $\rho_2$, with $C_{\rho_2} = C_2$, then $\rho_1 \cdot \rho_2$ is a transfer-exponent from $P$ to $R$, with $C_{\rho_1 \rho_2} = C_2 \cdot (C_1)^{\frac{1}{\rho_2}}$.*

### 5.1. The Case of Knowing $i^* = \arg\min_{i \in [K]} \rho_i$

In this section, we take a step toward the nearly optimal algorithm by greatly reducing the amount of distributional

information we need in the oracle algorithm. We will explore nearly optimal algorithms without any knowledge of distributional parameters in the next section.

The oracle algorithm needs distributional parameters $\rho_i, \beta_i, c_{\rho,i}$ for $i \in [K]$. In this section, we consider the case where we know $i^* = \arg\min_{i \in [K]} \rho_i$, which is much better than the case considered in the oracle algorithm. Without loss of generality, we assume that $1 = \arg\min_{i \in [K]} \rho_i$. To facilitate our algorithm design, we consider an alternative approach for modeling the relationships among the distributions.

**Definition 5.8** ($\mathcal{Q}$ Class). Let $\rho_1, \bar{\rho}_2, \ldots, \bar{\rho}_K \in (0, \infty], \beta_1, \ldots, \beta_K, \beta \in [0, 1], C > 0$. For a distribution tuple $(P_1, \ldots, P_K, Q)$, we define that $(P_1, \ldots, P_K, Q) \in \mathcal{Q}(\mathcal{H}; \rho_1, \bar{\rho}_2, \ldots, \bar{\rho}_K; \beta_1, \ldots, \beta_K; \beta; C)$ if the following conditions hold:

- For all $i \in [K]$, $P_i$ satisfies the RBCC with parameters $(\beta_i, c_i)$ with $c_i \leq C$. $Q$ satisfies the RBCC with parameters $(\beta, c)$ with $c \leq C$.

- For all $i \in [K] \backslash \{1\}$, the transfer exponent from $P_i$ to $P_1$ with respect to $\mathcal{H}$ is $\bar{\rho}_i$, with the corresponding constant $C_{\bar{\rho}_i} \leq C$. The transfer exponent from $P_1$ to $Q$ with respect to $\mathcal{H}$ is $\rho_1$, with the corresponding constant $C_{\rho_1} \leq C$.

When the context is clear, we denote $(P_1, \ldots, P_K, Q) \in \mathcal{Q}(\mathcal{H}; \rho_1, \bar{\rho}_2, \ldots, \bar{\rho}_K; \beta_1, \ldots, \beta_K; \beta; C)$ as $\mathcal{Q}$ for simplicity.

**Theorem 5.9** (Upper Bound for Algorithm 4). *Let $\mathcal{H}$ be a hypothesis class with $VCdim(\mathcal{H}) = d$. Let $(P_1, \ldots, P_K, Q) \in \mathcal{Q}(\mathcal{H}; \rho_1, \bar{\rho}_2, \ldots, \bar{\rho}_K; \beta_1, \ldots, \beta_K; \beta; C)$. If Assumption 5.5 holds and $\hat{h}$ is the output of Algorithm 4, then there*

*exists a constant $C' > 0$ (depending on $\rho_1, \bar{\rho}_i, \beta_i, C$) such that with probability at least $1 - \delta$,*

$$\mathcal{E}_Q\left(\hat{h}\right) \leq C' \cdot \min_{i \in [K]} \left\{ A_{n_i,K}^{\frac{1}{(2-\beta_i)\bar{\rho}_i\rho_1}} \right\},$$

*where we define $\bar{\rho}_1 = 1$ for simplicity.*

*Remark* 5.10 (Nearly Optimal Error Rate). If we set $\bar{\rho}_j = \frac{\rho_j}{\rho_1}$ for $j = 2, 3, \ldots, K$, then the class $\mathcal{Q}(\mathcal{H}; \rho_1, \bar{\rho}_2, \ldots, \bar{\rho}_K; \beta_1, \ldots, \beta_K; \beta; C)$ reduces to class $\mathcal{P}(\mathcal{H}; \rho_1, \ldots, \rho_K; \beta_1, \ldots, \beta_K; \beta; 0, \ldots, 0; C)$, which means that we can adapt the lower bound for $\mathcal{P}$ as a lower bound for $\mathcal{Q}$ (for a rigorous proof, please refer to the proof of Theorem 5.12, more precisely, the proof of Lemma B.4). Applying Theorem 4.2 with $\rho_j = \rho_1 \cdot \bar{\rho}_j$ for $j = 2, 3, \ldots, K$ and $\alpha_1 = \cdots = \alpha_K = 0$, we know that the lower bound for $\mathcal{Q}(\mathcal{H}; \rho_1, \bar{\rho}_2, \ldots, \bar{\rho}_K; \beta_1, \ldots, \beta_K; \beta; C)$ with $\rho_1 \geq 1, \bar{\rho}_2 \geq 1, \ldots, \bar{\rho}_K \geq 1$ is of order:

$$\Omega\left( \min\left\{ \left(\frac{d}{n_1}\right)^{\frac{1}{(2-\beta_1)\bar{\rho}_1\rho_1}}, \ldots, \left(\frac{d}{n_K}\right)^{\frac{1}{(2-\beta_K)\bar{\rho}_K\rho_1}} \right\} \right),$$

and the upper bound is of order

$$O\left( \min_{1 \leq i \leq K} \left\{ A_{n_i,K}^{\frac{1}{(2-\beta_i)\bar{\rho}_i\rho_1}} \right\} \right) = \tilde{O}\left( \min_{1 \leq i \leq K} \left\{ \left(\frac{d}{n_i}\right)^{\frac{1}{(2-\beta_i)\bar{\rho}_i\rho_1}} \right\} \right).$$

So the bounds here are **nearly optimal** (up to a logarithm term).

According to Remark 5.10, Algorithm 4 achieves a nearly optimal error rate. However, the algorithm requires the knowledge of $i^* = \arg\min_{i \in [K]} \rho_i$, which is not accessible in practice. In light of this, a natural question arises:

> *Can we achieve a (nearly) optimal error rate without knowing $i^*$?*

We will answer the question in the next section.

## 5.2. The Case of Not Knowing Distributional Parameters

In this section, we affirmatively answer the question raised in Section 5.1 by proposing a simple yet powerful algorithm that achieves a nearly optimal error rate without knowing $i^*$, with a mild assumption.

An important observation is that in Algorithm 4, the knowledge of $i^*$ is mainly used to determine which source domain to minimize the empirical risk on. To be specific, we use domains with indices $[K] \backslash \{i^*\}$ to construct a subset $\cap_{j \in [K], j \neq i^*} \mathcal{H}_j \subseteq \mathcal{H}$ and minimize the empirical risk on the $i^*$-th source domain with the constraint $h \in \cap_{j \in [K], j \neq i^*}^K \mathcal{H}_j$. If we can design an algorithm which outputs a hypothesis

---

**Algorithm 5** The ERM Algorithm

**Input:** $\mathcal{H}, \{S_{P_i}\}_{i=1}^K$.
Choose $\hat{h}_{\text{ERM}} \in \arg\min_{h \in \mathcal{H}} \frac{1}{K} \sum_{k=1}^K R_{\hat{P}_k}(h)$.
**Output:** $\hat{h}_{\text{ERM}}$.

---

that not only belongs to $\cap_{j \in [K], j \neq i^*} \mathcal{H}_j$, but also minimizes the empirical risk on the $i^*$-th source domain without knowing $i^*$, then we can achieve a nearly optimal error rate without knowing $i^*$. The following arguments show that under a mild empirical version of Assumption 5.5, we can design such an algorithm.

**Definition 5.11** (The Empirical RCS Assumption). Let $S = (S_{P_1}, \ldots, S_{P_K})$ be the training set with $K$ independent labeled samples $S_{P_1} \sim P_1^{n_1}, \ldots, S_{P_K} \sim P_K^{n_K}$. Let $\hat{\mathcal{H}}_i = \arg\min_{h \in \mathcal{H}} R_{\hat{P}_i}(h)$ for $i \in [K]$, we say that $(\mathcal{H}, S)$ satisfies the empirical RCS condition if $\cap_{i=1}^K \hat{\mathcal{H}}_i \neq \emptyset$, i.e., there exists some $\hat{h} \in \mathcal{H}$ such that $\hat{h} \in \arg\min_{h \in \mathcal{H}} R_{\hat{P}_i}(h)$ for $i \in [K]$.

Then we can design a simple algorithm that reaches a nearly optimal error rate when the RCS and empirical RCS conditions hold.

**Theorem 5.12** (Error Bounds for Algorithm 5). *Let $\mathcal{H}$ be a hypothesis class with $VCdim(\mathcal{H}) = d$. Fix any $\rho_1, \ldots, \rho_K \in [1, \infty], \beta_1, \ldots, \beta_K, \beta \in [0, 1], C > 0$. Let $\mathcal{Q}$ be the $\mathcal{Q}$-class related to $\mathcal{P}(\mathcal{H}; \rho_1, \ldots, \rho_K; \beta_1, \ldots, \beta_K; \beta; 0, \ldots, 0; C)$ (please refer to Definition B.2). Let $(P_1, \ldots, P_K, Q) \in \mathcal{Q}$. If Assumption 5.5 holds and $\hat{h}_{ERM}$ is the output of Algorithm 5, then there exists constant $c, C' > 0$ such that with probability at least $1 - \delta$, if $(\mathcal{H}, S)$ satisfies the empirical RCS condition,*

$$c \cdot \min_{i \in [K]} \left\{ \left(\frac{d}{n_i}\right)^{\frac{1}{(2-\beta_i)\rho_i}} \right\} \leq \mathcal{E}_Q\left(\hat{h}_{ERM}\right) \leq C' \cdot \min_{i \in [K]} \left\{ A_{n_i,K}^{\frac{1}{(2-\beta_i)\rho_i}} \right\},$$

*where $A_{n,K} := \frac{d}{n} \ln\left(\frac{\max\{n,d\}}{d}\right) + \frac{\ln K/(3\delta)}{n}$.*

Similar to Algorithm 4, Algorithm 5 also achieves a nearly optimal excess risk error rate (up to a logarithm term). Moreover, Algorithm 5 does not need the information of $i^* = \arg\min_{i \in [K]} \rho_i$, which means that **our lower and upper bounds are tight under the RCS and empirical-RCS conditions**.

*Remark* 5.13 (Scope of the ERM near-optimality result). In the deep learning regime where we use powerful neural networks as the hypothesis classes, we can easily get a $\hat{h}_{\text{ERM}}$ so that $\frac{1}{K} \sum_{i=1}^K R_{\hat{P}_i}\left(\hat{h}_{\text{ERM}}\right) = 0$. So $\hat{h}_{\text{ERM}}$ simultaneously

minimizes the empirical risk of all source domains, which means that the empirical RCS condition holds. However, this empirical fact alone does not imply the population-level RCS condition, nor does it guarantee that the same classifier is optimal on an unseen target domain. Therefore, the near-optimality conclusion of Theorem 4.12 relies on the combination of the population RCS assumption and the empirical RCS condition, rather than on interpolation alone.

For convenience, we denote $R_S(h) = \frac{1}{K} \sum_{i=1}^{K} R_{\hat{P}_i}(h)$. Since $\min_{h \in \mathcal{H}} R_S(h) = 0$ is a sufficient condition for the empirical RCS condition, we can get an optimistic-type bound (Srebro et al., 2010) for Algorithm 5, where a zero empirical risk helps get a better error rate.

**Corollary 5.14** (Optimistic-Type Bound for Algorithm 5). *Let $\mathcal{H}$ be a hypothesis class with $VCdim(\mathcal{H}) = d$. Fix any $\rho_1, \ldots, \rho_K \in [1, \infty], \beta_1, \ldots, \beta_K, \beta \in [0, 1], C > 0$. Let $\mathcal{Q}$ be the $\mathcal{Q}$-class related to $\mathcal{P}(\mathcal{H}; \rho_1, \ldots, \rho_K; \beta_1, \ldots, \beta_K; \beta; 0, \ldots, 0; C)$. Let $(P_1, \ldots, P_K, Q) \in \mathcal{Q}$. If Assumption 5.5 holds and $\hat{h}_{ERM}$ is the output of Algorithm 5, then there exists constant $C_1, C_2 > 0$ such that with probability at least $1 - \delta$,*

$$\mathcal{E}_Q\left(\hat{h}_{ERM}\right) \leq C_1 \cdot \mathbb{1}\left[R_S\left(\hat{h}_{ERM}\right) = 0\right] \cdot \min_{i \in [K]}\left\{A_{n_j, K}^{\frac{1}{(2-\beta_j)\rho_j}}\right\}$$
$$+ C_2 \cdot \mathbb{1}\left[R_S\left(\hat{h}_{ERM}\right) > 0\right] \cdot \max_{i \in [K]}\left\{A_{n_j, K}^{\frac{1}{(2-\beta_j)\rho_j}}\right\}.$$

*Remark* 5.15 (A conditional perspective on the phenomenon in DomainBed (Gulrajani & Lopez-Paz, 2021)). In DomainBed, the authors conducted experiments on many OOD generalization methods and found that the ERM algorithm (Vapnik, 1991) shows state-of-the-art performance across all datasets. The result is surprising since the simple ERM algorithm, which is not specifically designed for OOD generalization, is not worse than other carefully designed OOD generalization algorithms. The ERM algorithm in DomainBed outputs a hypothesis $\hat{h}_{ERM} = \arg\min_{h \in \mathcal{H}} \frac{1}{K} \sum_{k=1}^{K} R_{\hat{P}_k}(h) = \arg\min_{h \in \mathcal{H}} R_S\left(\hat{h}\right)$, which is exactly the output of Algorithm 5. Therefore, our result offers one conditional theoretical perspective on why ERM can be competitive when the RCS / empirical-RCS regime approximately captures the structure of the problem.

## 6. Discussion

In this section, we make discussions on the relationships among the transfer-exponent, our RCS assumption, and other assumptions (the commonly used covariate shift and concept shift) that characterize the distributional shift. We also point out possible future works.

**Transfer-exponent & other distributional shift assumptions.** In fact, unlike the covariate shift and concept shift

assumptions, the transfer-exponent is not an assumption. It is a way of characterizing the knowledge transfer ability between two distributions, which also depends on the choice of the hypothesis class $\mathcal{H}$. For any two distributions, there always exists a $\rho > 0$ such that $\rho$ is the transfer-exponent from $P$ to $Q$. So, transfer-exponent is not exclusive of other well-studied distributional shift assumptions. For example, the constructed distributional tuples in the proof of our lower bound satisfy the concept shift assumption.

**RCS assumption & other distributional shift assumptions.** Our main assumption for the upper bound is the RCS assumption. It is useful as it serves to isolate the sources of hardness in transfer beyond just shifts in $h_P^*$ and $h_Q^*$, so it is still worth studying. As stated in (Hanneke & Kpotufe, 2019), RCS assumption is neither strictly weaker nor strictly stronger than the well-studied covariate shift assumption and concept shift assumption. More concretely, the RCS assumption allows $P_{Y|X} \neq Q_{Y|X}$, so it is not strictly stronger than the covariate shift assumption. However, when the Bayes classifier is not in $\mathcal{H}$, covariate shift allows $h_P^* \neq h_Q^*$, which means that the RCS assumption is not strictly weaker than the covariate shift assumption. As for the concept shift, RCS allows $P_X \neq Q_X$, so it is not strictly stronger than the concept shift assumption. When the Bayes classifier $\in \mathcal{H}$, it is obvious that concept shift implies $h_P^* \neq h_Q^*$, so the RCS assumption is not strictly weaker than the concept shift assumption.

**A hierarchical Bayesian perspective.** It may be worthwhile to explore whether the upper-bound regime could be interpreted through a hierarchical Bayesian perspective, which might soften the exact one-optimum-hypothesis assumptions and make the framework more practically relevant. Such a perspective could more naturally model partial sharing across domains than the current formulation.

## 7. Conclusion

In this paper, we investigate upper and lower bounds on out-of-distribution (OOD) generalization from the perspective of statistical learning. Our bounds show that, under the RCS and empirical-RCS conditions, minimizing the average empirical risk over the source domains is nearly optimal up to logarithmic factors. This provides a conditional theoretical perspective on the competitive performance of ERM observed in DomainBed. Furthermore, our results allow us to derive a no-free-lunch theorem and an optimistic bound for OOD generalization.

## Acknowledgments and Disclosure of Funding

This work is supported by the National Natural Science Foundation of China under Grant 624B2106, the Key R&D Program of Hubei Province under Grant 2024BAB038, and

the National Key R&D Program of China under Grant 2023YFC3604702.

## Impact Statement

This paper presents work whose goal is to advance the field of Machine Learning. There are many potential societal consequences of our work, none which we feel must be specifically highlighted here.

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

# A. Proofs of the Lower Bounds

**Definition A.1** (Kullback-Leibler divergence). If $P$ and $Q$ are two probability measures on a measurable space $\mathcal{X}$, and $P$ is absolutely continuous with respect to $Q$, then the **Kullback-Leibler divergence** from $Q$ to $P$ is defined as

$$D_{\mathrm{KL}}(P\|Q) := \int_{x\in\mathcal{X}} \log\left(\frac{dP}{dQ}(x)\right) dP(x),$$

where $\frac{dP}{dQ}$ is the **Radon-Nikodym derivative** (Patrick, 2008) of $P$ with respect to $Q$.

**Lemma A.2** (Some Properties of the Kullback-Leibler Divergence). *Assume that the probability density functions of $P, P_1, P_2$ and $Q, Q_1, Q_2$ exists, and are denoted as $p, p_1, p_2$ and $q, q_1, q_2$ respectively. Then, the following properties hold:*

1. *If $P_1 \ll P_2$ and $Q_1 \ll Q_2$, where $\ll$ denotes absolute continuity, then $D_{KL}(P_1 \times Q_1 \| P_2 \times Q_2) = D_{KL}(P_1 \| P_2) + D_{KL}(Q_1 \| Q_2)$.*

2. *If $P \ll Q$, then $D_{KL}(P^n \| Q^n) = n \cdot D_{KL}(P \| Q)$.*

3. *If $P \ll Q$, and $P_X = Q_X$ then $D_{KL}(P \| Q) = \underset{X \sim P_X}{\mathbb{E}} D_{KL}(P_{Y|X} \| Q_{Y|X})$*

*Proof of Lemma A.2.* We first prove the first property.

$$\begin{aligned}
D_{\mathrm{KL}}(P_1 \times Q_1 \| P_2 \times Q_2) &= \int_{x\in\mathcal{X}} \int_{y\in\mathcal{X}} \log\left(\frac{d(P_1\times Q_1)}{d(P_2\times Q_2)}(x,y)\right) dP_1(x)dQ_1(y) \\
&= \int_{x\in\mathcal{X}} \int_{y\in\mathcal{X}} p_1(x)q_1(y) \log\left(\frac{p_1(x)q_1(y)}{p_2(x)q_2(y)}\right) dxdy \\
&= \int_{x\in\mathcal{X}} \int_{y\in\mathcal{X}} p_1(x)q_1(y) \log\left(\frac{p_1(x)}{p_2(x)}\right) dxdy + \int_{x\in\mathcal{X}} \int_{y\in\mathcal{X}} p_1(x)q_1(y) \log\left(\frac{q_1(y)}{q_2(y)}\right) dxdy \\
&= \int_{x\in\mathcal{X}} p_1(x) \log\left(\frac{p_1(x)}{p_2(x)}\right) dx + \int_{y\in\mathcal{X}} q_1(y) \log\left(\frac{q_1(y)}{q_2(y)}\right) dy \\
&= D_{\mathrm{KL}}(P_1\|P_2) + D_{\mathrm{KL}}(Q_1\|Q_2).
\end{aligned}$$

The second property can be proved by repeatedly applying the first property. Then we prove the third property.

$$\begin{aligned}
D_{\mathrm{KL}}(P\|Q) &= \int_{x\in\mathcal{X},y\in\mathcal{Y}} p(x,y) \log\left(\frac{p(x,y)}{q(x,y)}\right) dxdy \\
&= \int_{x\in\mathcal{X},y\in\mathcal{Y}} p(x)p(y|x) \log\left(\frac{p(x)p(y|x)}{q(x)q(y|x)}\right) dxdy \\
&\overset{a}{=} \int_{x\in\mathcal{X},y\in\mathcal{Y}} p(x)p(y|x) \log\left(\frac{p(y|x)}{q(y|x)}\right) dxdy \\
&= \int_{x\in\mathcal{X}} p(x) \left(\int_{y\in\mathcal{Y}} p(y|x) \log\left(\frac{p(y|x)}{q(y|x)}\right) dy\right) dx \\
&= \int_{x\in\mathcal{X}} p(x) D_{\mathrm{KL}}(P_{Y|X=x}\|Q_{Y|X=x})dx \\
&= \underset{X\sim P_X}{\mathbb{E}} D_{\mathrm{KL}}(P_{Y|X}\|Q_{Y|X}),
\end{aligned}$$

where $(a)$ follows from the fact that $P_X = Q_X$. $\qquad\square$

Our lower bound relies on the following extension of the Fano's inequality.

**Lemma A.3** (Theorem 2.5 of (Tsybakov, 2009)). *Let $\{\Pi_{\boldsymbol{\sigma}}\}_{\boldsymbol{\sigma}\in\Sigma}$ be a family of distributions indexed over a subset $\Sigma$ of a semi-metric $(\mathcal{F}, H)$. Suppose $\exists \boldsymbol{\sigma}_0, \ldots, \boldsymbol{\sigma}_M \in \Sigma$, where $M \geq 2$, such that:*

1. *$H(\boldsymbol{\sigma}_i, \boldsymbol{\sigma}_j) \geq 2s > 0, \ \forall 0 \leq i < j \leq M$,*

2. $\Pi_{\boldsymbol{\sigma}_i} \ll \Pi_{\boldsymbol{\sigma}_0}$, and the average KL-divergence to $\Pi_{\boldsymbol{\sigma}_0}$ satisfies

$$\frac{1}{M} \sum_{i=1}^{M} D_{KL}(\Pi_{\boldsymbol{\sigma}_i} \| \Pi_{\boldsymbol{\sigma}_0}) \leq \alpha \log M, \quad where \ 0 < \alpha < \frac{1}{8}.$$

Let $Z \sim \Pi_{\boldsymbol{\sigma}}$, and let $\hat{h} : Z \to \mathcal{F}$ denote any improper learner of $\boldsymbol{\sigma} \in \Sigma$. We have that for any $\hat{h}$:

$$\sup_{\boldsymbol{\sigma} \in \Sigma} \mathop{\mathbb{P}}_{Z \sim \Pi_{\boldsymbol{\sigma}}} \left[ H\left(\hat{h}(Z), \boldsymbol{\sigma}\right) \geq s \right] \geq \frac{\sqrt{M}}{1 + \sqrt{M}} \left(1 - 2\alpha - \sqrt{\frac{2\alpha}{\log M}}\right) \geq \frac{3 - 2\sqrt{2}}{8}.$$

The following lemma would be needed to construct packings (of spaces of distributions) of the appropriate size.

**Lemma A.4** (Varshamov-Gilbert bound, Proposition 5 in (Hanneke & Kpotufe, 2019)). *Let $d \geq 8$. Then there exists a subset $\{\boldsymbol{\sigma}_0, \ldots, \boldsymbol{\sigma}_M\}$ of $\{-1, +1\}^d$ such that $\boldsymbol{\sigma}_0 = (1, \ldots, 1)$,*

$$H(\boldsymbol{\sigma}_i, \boldsymbol{\sigma}_j) \geq \frac{d}{8}, \quad \forall 0 \leq i < j \leq M, \quad and \quad M \geq 2^{\frac{d}{8}},$$

*where $H(\cdot, \cdot)$ is the Hamming distance.*

**Lemma A.5** (A basic KL upper bound for Bernoulli distributions, Lemma 2 in (Hanneke & Kpotufe, 2019)). *Let $p, q \in (0, 1)$, we denote $D_{KL}\left(\boldsymbol{Bernoulli}(p) \| \boldsymbol{Bernoulli}(q)\right)$ as $D_{KL}\left(p \| q\right)$ for short. Then, there exists a universal constant $c_0 > 0$ such that for any $\epsilon \in (0, \frac{1}{2})$ and any $\sigma \in \{-1, +1\}$ such that*

$$D_{KL} \left(\frac{1}{2} + \frac{\sigma}{2} \cdot \epsilon \middle\| \frac{1}{2} - \frac{\sigma}{2} \cdot \epsilon \right) \leq c_0 \cdot \epsilon^2.$$

*Proof of Theorem 4.2.* For simplicity, for any $i \in [K]$, we define

$$\varepsilon_i = \left(\frac{d}{n_i}\right)^{\frac{1}{(2 - \beta_i)\rho_i}}.$$

Let

$$s_K = \min \{\varepsilon_1 + \alpha_1, \ldots, \varepsilon_K + \alpha_K\}.$$

For any $j \in [K]$, we define $r_j = \max\{\varepsilon_j, \alpha_j\}$, and then define

$$s'_K = \min \{r_1, \ldots, r_K\}.$$

It is easy to verify that

$$s'_K \leq s_K \leq 2s'_K.$$

By the assumption on that

$$\min \left\{ \left(\frac{d}{n_1}\right)^{\frac{1}{(2 - \frac{1}{\beta_1})\rho_1}}, \ldots, \left(\frac{d}{n_K}\right)^{\frac{1}{(2 - \frac{1}{\beta_K})\rho_K}} \right\} = \min \{\varepsilon_1, \ldots, \varepsilon_K\} \leq 1,$$

we know that there exists one $j \in [K]$ such that $\varepsilon_j \leq 1$. Since $\alpha_j \leq 1$, we know that $r_j \leq 1$. Therefore, we have $s'_K \leq 1$. Fix $s = c_1 \cdot s'_K$ where $c_1 \in (0, 1)$ is a constant to be determined later, we have $s < c_1 < 1$ (which means that the distributions defined below are valid probability Distributions).

Since $d = \text{VCdim}(\mathcal{H})$ is the Vapnik-Chervonenkis dimension of $\mathcal{H}$, there exists a subset $\mathcal{S} = \{x_0, x_1, \ldots, x_{d-1}\} \subseteq \mathcal{X}$ that is shattered by $\mathcal{H}$. Let $\tilde{\mathcal{H}}$ be the projection of $\mathcal{H}$ onto the set $\mathcal{S}$, and define the "Hamming distance" on $\tilde{\mathcal{H}}$ as:

$$H(h, h') = \sum_{i=1}^{d-1} \mathbb{1}[h(x_i) \neq h'(x_i)], \quad \forall h, h' \in \tilde{\mathcal{H}}.$$

It is easy to verify that $H$ satisfies the conditions for a semi-metric. Let $\boldsymbol{\sigma} \in \{-1, +1\}^{d-1}$, we will construct a family of distribution tuples $(P_{1,\boldsymbol{\sigma}}, \cdots, P_{K,\boldsymbol{\sigma}}, Q_{\boldsymbol{\sigma}})$ indexed by $\boldsymbol{\sigma} \in \{-1, +1\}^{d-1}$ and then apply Lemma A.3 to prove the lower bound. For simplicity, we use $h_{\boldsymbol{\sigma}}$ to present the hypothesis in $\tilde{\mathcal{H}}$ that where $h_{\boldsymbol{\sigma}}(x_0) = 1$ and $h_{\boldsymbol{\sigma}}(x_i) = \boldsymbol{\sigma}_i$ for $i \in [d-1]$; and use $h'_{\boldsymbol{\sigma}}$ to present the hypothesis in $\tilde{\mathcal{H}}$ that where $h'_{\boldsymbol{\sigma}}(x_0) = -1$ and $h'_{\boldsymbol{\sigma}}(x_i) = \boldsymbol{\sigma}_i$ for $i \in [d-1]$. Let $\boldsymbol{\eta}_1, \ldots, \boldsymbol{\eta}_K \in \{-1, +1\}^{d-1}$ be $K$ pairwise distinct (not necessarily distinct) vectors, which is valid if $K \le 2^{d-1}$.

**Construction of the Distribution $Q_{\boldsymbol{\sigma}}$.** Let $Q_{\boldsymbol{\sigma}} = Q_X \times Q_{\boldsymbol{\sigma}, Y|X}$. We set $Q_X(x_0) = 1 - s^{\beta}$ and $Q_X(x_i) = \frac{s^{\beta}}{d-1}$ for $i \in [d-1]$. For $Q_{\boldsymbol{\sigma}, Y|X}$, we set $Q_{\boldsymbol{\sigma}, Y|X=x_0}(Y=1) = \frac{1}{2}$ and $Q_{\boldsymbol{\sigma}, Y|X=x_i}(Y=1) = \frac{1}{2} + \frac{\boldsymbol{\sigma}_i}{2} \cdot s^{1-\beta}$ for $i \in [d-1]$.

**Construction of the Distribution $P_{j,\boldsymbol{\sigma}}$ for $j \in [K]$.** The construction of $P_{j,\boldsymbol{\sigma}}$ depends on the exact value of $r_j$. We consider the following two cases:

- If $r_j = \varepsilon_j$, we set $P_{j,\boldsymbol{\sigma}} = P_{j,X} \times P_{j,\boldsymbol{\sigma}, Y|X}$. We set $P_{j,X}(x_0) = 1 - s^{\rho_j \beta_j}$ and $P_{j,X}(x_i) = \frac{s^{\rho_j \beta_j}}{d-1}$ for $i \in [d-1]$. For $P_{j,\boldsymbol{\sigma}, Y|X}$, we set $P_{j,\boldsymbol{\sigma}, Y|X=x_0}(Y=1) = \frac{1}{2}$ and $P_{j,\boldsymbol{\sigma}, Y|X=x_i}(Y=1) = \frac{1}{2} + \frac{\boldsymbol{\sigma}_i}{2} \cdot s^{\rho_j(1-\beta_j)}$ for $i \in [d-1]$.

- If $r_j = \alpha_j$, we set $P_{j,\boldsymbol{\sigma}} = P'_j = P'_{j,X} \times P'_{j,Y|X}$. We set $P'_{j,X}(x_0) = 0$ and $P'_{j,X}(x_i) = \frac{1}{d-1}$ for $i \in [d-1]$. For $P'_{j,Y|X}$, we set $P_{j,Y|X=x_i}(Y = \boldsymbol{\eta}_{j,i}) = 1$.

We then verify that the tuples $\{(P_{1,\boldsymbol{\sigma}}, \cdots, P_{K,\boldsymbol{\sigma}}, Q_{\boldsymbol{\sigma}})\}_{\boldsymbol{\sigma} \in \{-1,+1\}^{d-1}}$ belongs to the class $\mathcal{P}(\mathcal{H}; \rho_1, \ldots, \rho_K; \beta_1, \ldots, \beta_K; \beta; \alpha_1, \ldots, \alpha_K; 1)$. We first analyze the distributions.

**Analysis of the Distributions $Q_{\boldsymbol{\sigma}}$.** From the construction of $Q_{\boldsymbol{\sigma}}$, we know that: $\{h_{\boldsymbol{\sigma}}, h'_{\boldsymbol{\sigma}}\} = \arg\min_{h \in \tilde{\mathcal{H}}} R_{Q_{\boldsymbol{\sigma}}}(h)$ are the Bayes classifiers of $Q_{\boldsymbol{\sigma}}$. Then, for any $\boldsymbol{\sigma}, \boldsymbol{\sigma}' \in \{-1, +1\}^{d-1}$ we have:

$$\mathcal{E}_{Q_{\boldsymbol{\sigma}}}(h_{\boldsymbol{\sigma}'}) = \mathcal{E}_{Q_{\boldsymbol{\sigma}}}(h'_{\boldsymbol{\sigma}'}) = H(\boldsymbol{\sigma}, \boldsymbol{\sigma}') \cdot \frac{s^{\beta}}{d-1} \cdot s^{1-\beta} = \frac{H(\boldsymbol{\sigma}, \boldsymbol{\sigma}')}{d-1} \cdot s, \tag{4}$$

$$Q_{\boldsymbol{\sigma}}(h_{\boldsymbol{\sigma}'} \ne h_{\boldsymbol{\sigma}}) = Q_{\boldsymbol{\sigma}}(h'_{\boldsymbol{\sigma}'} \ne h'_{\boldsymbol{\sigma}}) = H(\boldsymbol{\sigma}, \boldsymbol{\sigma}') \cdot \frac{s^{\beta}}{d-1} = \frac{H(\boldsymbol{\sigma}, \boldsymbol{\sigma}')}{d-1} \cdot s^{\beta}. \tag{5}$$

**Analysis of the Distributions $P_{j,\boldsymbol{\sigma}} = P_{j,X} \times P_{j,\boldsymbol{\sigma}|X}$.** From the construction of $P_{j,\boldsymbol{\sigma}}$, we know that: $\{h_{\boldsymbol{\sigma}}, h'_{\boldsymbol{\sigma}}\} = \arg\min_{h \in \tilde{\mathcal{H}}} R_{P_{j,\boldsymbol{\sigma}}}(h)$ are the Bayes classifiers of $P_{j,\boldsymbol{\sigma}}$ for $j \in [K]$. Then, for any $\boldsymbol{\sigma}, \boldsymbol{\sigma}' \in \{-1, +1\}^{d-1}$ we have:

$$\mathcal{E}_{P_{j,\boldsymbol{\sigma}}}(h_{\boldsymbol{\sigma}'}) = \mathcal{E}_{P_{j,\boldsymbol{\sigma}}}(h'_{\boldsymbol{\sigma}'}) = H(\boldsymbol{\sigma}, \boldsymbol{\sigma}') \cdot \frac{s^{\rho_j \beta_j}}{d-1} \cdot s^{\rho_j(1-\beta_j)} = \frac{H(\boldsymbol{\sigma}, \boldsymbol{\sigma}')}{d-1} \cdot s^{\rho_j}, \tag{6}$$

$$P_{j,\boldsymbol{\sigma}}(h_{\boldsymbol{\sigma}'} \ne h_{\boldsymbol{\sigma}}) = P_{j,\boldsymbol{\sigma}}(h'_{\boldsymbol{\sigma}'} \ne h'_{\boldsymbol{\sigma}}) = H(\boldsymbol{\sigma}, \boldsymbol{\sigma}') \cdot \frac{s^{\rho_j \beta_j}}{d-1} = \frac{H(\boldsymbol{\sigma}, \boldsymbol{\sigma}')}{d-1} \cdot s^{\rho_j \beta_j}. \tag{7}$$

**Analysis of the Distributions $P_{j,\boldsymbol{\sigma}} = P'_j$.** From the construction of $P'_j$, we know that: $\{h_{\boldsymbol{\eta}_j}, h'_{\boldsymbol{\eta}_j}\} = \arg\min_{h \in \tilde{\mathcal{H}}} R_{P'_j}(h)$ are the Bayes classifiers of $P'_j$ for $j \in [K]$. Furthermore, $R_{P'_j}(h_{\boldsymbol{\eta}_j}) = R_{P'_j}(h'_{\boldsymbol{\eta}_j}) = 0$ for any $j \in [K]$. Then, for any $\boldsymbol{\sigma} \in \{-1, +1\}^{d-1}$ we have:

$$\mathcal{E}_{P'_j}(h_{\boldsymbol{\sigma}}) = \mathcal{E}_{P'_j}(h'_{\boldsymbol{\sigma}}) = H(\boldsymbol{\eta}_j, \boldsymbol{\sigma}) \cdot \frac{1}{d-1} \cdot 1 = \frac{H(\boldsymbol{\eta}_j, \boldsymbol{\sigma})}{d-1}, \tag{8}$$

$$P'_j(h_{\boldsymbol{\sigma}} \ne h_{\boldsymbol{\eta}_j}) = P'_j(h'_{\boldsymbol{\sigma}} \ne h'_{\boldsymbol{\eta}_j}) = H(\boldsymbol{\eta}_j, \boldsymbol{\sigma}) \cdot \frac{1}{d-1} = \frac{H(\boldsymbol{\eta}_j, \boldsymbol{\sigma})}{d-1}. \tag{9}$$

Now we are ready to verify $(P_{1,\boldsymbol{\sigma}}, \cdots, P_{K,\boldsymbol{\sigma}}, Q_{\boldsymbol{\sigma}}) \in \mathcal{P} \ \forall \boldsymbol{\sigma} \in \{-1, +1\}^{d-1}$.

**Verification of RBCC.**

- Regarding $Q_{\boldsymbol{\sigma}}$, for any $h_{\boldsymbol{\sigma}'} \in \tilde{\mathcal{H}}$, we have:

$$Q_{\boldsymbol{\sigma}}(h_{\boldsymbol{\sigma}'} \neq h_{\boldsymbol{\sigma}}) = \frac{H(\boldsymbol{\sigma}, \boldsymbol{\sigma}')}{d-1} \cdot s^{\beta}$$

$$= \left( \frac{H(\boldsymbol{\sigma}, \boldsymbol{\sigma}')}{d-1} \right)^{1-\beta} \cdot \left( \frac{H(\boldsymbol{\sigma}, \boldsymbol{\sigma}')}{d-1} \cdot s \right)^{\beta}$$

$$\overset{a}{\leq} \left( \frac{H(\boldsymbol{\sigma}, \boldsymbol{\sigma}')}{d-1} \cdot s \right)^{\beta} = \mathcal{E}_{Q_{\boldsymbol{\sigma}}}^{\beta}(h_{\boldsymbol{\sigma}'}),$$

where step $a$ follows from the fact that $x^{1-\beta} \leq 1$ for $x \in [0,1]$ and $\beta \in [0,1]$. Similarly, we can verify that $Q_{\boldsymbol{\sigma}}(h'_{\boldsymbol{\sigma}'} \neq h'_{\boldsymbol{\sigma}}) \leq \mathcal{E}_{Q_{\boldsymbol{\sigma}}}^{\beta}(h'_{\boldsymbol{\sigma}'})$ for $h'_{\boldsymbol{\sigma}'} \in \tilde{\mathcal{H}}$, which means that

$$\inf_{h^* \in \tilde{\mathcal{H}}_{Q_{\boldsymbol{\sigma}}}} Q_{\boldsymbol{\sigma}}(h \neq h^*) \leq \mathcal{E}_{Q_{\boldsymbol{\sigma}}}^{\beta}(h), \ \ \forall h \in \tilde{\mathcal{H}}.$$

So $Q_{\boldsymbol{\sigma}}$ satisfies the RBCC with parameters $(\beta, 1)$ for all $\boldsymbol{\sigma} \in \{-1, +1\}^{d-1}$.

- Regarding $P_{j,\boldsymbol{\sigma}} = P_{j,X} \times P_{j,\boldsymbol{\sigma},Y|X}$ for $j \in [K]$, for any $h_{\boldsymbol{\sigma}'} \in \tilde{\mathcal{H}}$, we have:

$$P_{j,\boldsymbol{\sigma}}(h_{\boldsymbol{\sigma}'} \neq h_{\boldsymbol{\sigma}}) = \frac{H(\boldsymbol{\sigma}, \boldsymbol{\sigma}')}{d-1} \cdot s^{\rho_j \beta_j}$$

$$= \left( \frac{H(\boldsymbol{\sigma}, \boldsymbol{\sigma}')}{d-1} \right)^{1-\beta_j} \cdot \left( \frac{H(\boldsymbol{\sigma}, \boldsymbol{\sigma}')}{d-1} \cdot s^{\rho_j} \right)^{\beta_j}$$

$$\overset{a}{\leq} \left( \frac{H(\boldsymbol{\sigma}, \boldsymbol{\sigma}')}{d-1} \cdot s^{\rho_j} \right)^{\beta_j} = \mathcal{E}_{P_{j,\boldsymbol{\sigma}}}^{\beta_j}(h_{\boldsymbol{\sigma}'}),$$

where step $a$ follows from the fact that $x^{1-\beta_j} \leq 1$ for $x \in [0,1]$ and $\beta_j \in [0,1]$. Similarly, we can verify that $P_{j,\boldsymbol{\sigma}}(h'_{\boldsymbol{\sigma}'} \neq h'_{\boldsymbol{\sigma}}) \leq \mathcal{E}_{P_{j,\boldsymbol{\sigma}}}^{\beta_j}(h'_{\boldsymbol{\sigma}'})$ for $h'_{\boldsymbol{\sigma}'} \in \tilde{\mathcal{H}}$, which means that

$$\inf_{h^* \in \tilde{\mathcal{H}}_{P_{j,\boldsymbol{\sigma}}}} P_{j,\boldsymbol{\sigma}}(h \neq h^*) \leq \mathcal{E}_{P_{j,\boldsymbol{\sigma}}}^{\beta_j}(h), \ \ \forall h \in \tilde{\mathcal{H}}.$$

So $P_{j,\boldsymbol{\sigma}}$ satisfies the RBCC with parameters $(\beta_j, 1)$ for all $\boldsymbol{\sigma} \in \{-1, +1\}^{d-1}$ and $j \in [K]$.

- Regarding $P'_j$ for $j \in [K]$, for any $h_{\boldsymbol{\sigma}} \in \tilde{\mathcal{H}}$, we have:

$$P'_j(h_{\boldsymbol{\sigma}} \neq h_{\boldsymbol{\eta}_j}) = \frac{H(\boldsymbol{\eta}_j, \boldsymbol{\sigma})}{d-1} = \mathcal{E}_{P'_j}(h_{\boldsymbol{\sigma}}) \overset{a}{\leq} \mathcal{E}_{P'_j}^{\beta_j}(h_{\boldsymbol{\sigma}}),$$

where step $a$ follows from the fact that $x \leq x^{\beta_j}$ for $x \in [0,1]$ and $\beta_j \in [0,1]$. Similarly, we can verify that $P'_j(h'_{\boldsymbol{\sigma}} \neq h'_{\boldsymbol{\eta}_j}) \leq \mathcal{E}_{P'_j}^{\beta_j}(h'_{\boldsymbol{\sigma}})$ for $h'_{\boldsymbol{\sigma}} \in \tilde{\mathcal{H}}$, which means that

$$\inf_{h^* \in \tilde{\mathcal{H}}_{P'_j}} P'_j(h \neq h^*) \leq \mathcal{E}_{P'_j}^{\beta_j}(h), \ \ \forall h \in \tilde{\mathcal{H}}.$$

So $P'_j$ satisfies the RBCC with parameters $(\beta_j, 1)$ for all $j \in [K]$.

**Verification of Transfer-Exponent.**

- Regarding transfer-exponent from $P_{j,\boldsymbol{\sigma}}$ to $Q_{\boldsymbol{\sigma}}$ where $P_{j,\boldsymbol{\sigma}} = P_{j,X} \times P_{j,\boldsymbol{\sigma},Y|X}$. Since $\arg\min_{h \in \tilde{\mathcal{H}}} R_{Q_{\boldsymbol{\sigma}}}(h) = \arg\min_{h \in \tilde{\mathcal{H}}} R_{P_{j,\boldsymbol{\sigma}}}(h)$ for any $j \in [K]$ and $\boldsymbol{\sigma} \in \{-1, +1\}^{d-1}$, we have $\mathcal{E}_{Q_{\boldsymbol{\sigma}}}\left( h^*_{P_{j,\boldsymbol{\sigma}}} \right) = 0$ for any $j \in [K]$ and $\boldsymbol{\sigma} \in \{-1, +1\}^{d-1}$. So for any $h_{\boldsymbol{\sigma}'}, h'_{\boldsymbol{\sigma}'} \in \tilde{\mathcal{H}}$, we have:

$$\mathcal{E}_{Q_{\boldsymbol{\sigma}}}\left( h_{\boldsymbol{\sigma}'}, h^*_{P_{j,\boldsymbol{\sigma}}} \right) = \mathcal{E}_{Q_{\boldsymbol{\sigma}}}\left( h'_{\boldsymbol{\sigma}'}, h^*_{P_{j,\boldsymbol{\sigma}}} \right) = \frac{H(\boldsymbol{\sigma}, \boldsymbol{\sigma}')}{d-1} \cdot s,$$

$$\mathcal{E}_{P_{j,\sigma}}\left(h_{\sigma'}, h^*_{P_{j,\sigma}}\right) = \mathcal{E}_{P_{j,\sigma}}\left(h'_{\sigma'}, h^*_{P_{j,\sigma}}\right) = \frac{H(\sigma, \sigma')}{d-1} \cdot s^{\rho_j},$$

which means that:

$$\begin{aligned}
\mathcal{E}_{Q_\sigma}\left(h_{\sigma'}, h^*_{P_{j,\sigma}}\right) &= \frac{H(\sigma, \sigma')}{d-1} \cdot s \\
&= \left(\frac{H(\sigma, \sigma')}{d-1}\right)^{1-\frac{1}{\rho_j}} \cdot \left(\frac{H(\sigma, \sigma')}{d-1} \cdot s^{\rho_j}\right)^{\frac{1}{\rho_j}} \\
&\overset{a}{\leq} \left(\frac{H(\sigma, \sigma')}{d-1} \cdot s^{\rho_j}\right)^{\frac{1}{\rho_j}} = \mathcal{E}_{P_{j,\sigma}}^{\frac{1}{\rho_j}}(h_{\sigma'}, h^*_{P_{j,\sigma}}),
\end{aligned}$$

where step $a$ follows from the fact that $x^{1-\frac{1}{\rho_j}} \leq 1$ for $x \in [0,1]$ and $\frac{1}{\rho_j} \in [0,1]$. Similarly, we can verify that $\mathcal{E}_{Q_\sigma}\left(h'_{\sigma'}, h^*_{P_{j,\sigma}}\right) \leq \mathcal{E}_{P_{j,\sigma}}^{\frac{1}{\rho_j}}(h'_{\sigma'}, h^*_{P_{j,\sigma}})$ for $h'_{\sigma'} \in \tilde{\mathcal{H}}$, which means that

$$\mathcal{E}_{Q_\sigma}\left(h, h^*_{P_{j,\sigma}}\right) \leq \mathcal{E}_{P_{j,\sigma}}^{\frac{1}{\rho_j}}(h), \quad \forall h \in \tilde{\mathcal{H}}.$$

So $P_{j,\sigma} \to Q_\sigma$ satisfies the transfer-exponent with parameters $(\rho_j, 1)$ for all $\sigma \in \{-1, +1\}^{d-1}$ and $j \in [K]$.

- Regarding transfer-exponent from $P'_j$ to $Q_\sigma$. Since $\{h_{\eta_j}, h'_{\eta_j}\} = \underset{h \in \tilde{\mathcal{H}}}{\arg\min}\, R_{P'_j}(h)$ for any $j \in [K]$, for any $\sigma' \in \{-1, +1\}^{d-1}$, we have:

$$\mathcal{E}_{P'_j}\left(h_{\sigma'}, h^*_{P'_j}\right) = \mathcal{E}_{P'_j}\left(h'_{\sigma'}, h^*_{P'_j}\right) = \frac{H(\eta_j, \sigma')}{d-1}.$$

For any $\sigma \in \{-1, +1\}^{d-1}$, we have:

$$\begin{aligned}
\mathcal{E}_{Q_\sigma}\left(h_{\sigma'}, h^*_{P'_j}\right) &= \mathcal{E}_{Q_\sigma}(h_{\sigma'}) - \mathcal{E}_{Q_\sigma}\left(h_{\eta_j}\right) \\
&= \frac{H(\sigma, \sigma')}{d-1} \cdot s - \frac{H(\sigma, \eta_j)}{d-1} \cdot s \\
&\overset{a}{\leq} \frac{H(\eta_j, \sigma')}{d-1} \cdot s \\
&= \left(\frac{H(\eta_j, \sigma')}{d-1}\right)^{1-\frac{1}{\rho_j}} \cdot \left(\frac{H(\eta_j, \sigma')}{d-1}\right)^{\frac{1}{\rho_j}} \cdot s \\
&\overset{b}{\leq} \left(\frac{H(\eta_j, \sigma')}{d-1}\right)^{\frac{1}{\rho_j}} = \mathcal{E}_{P'_j}^{\frac{1}{\rho_j}}(h_{\sigma'}, h^*_{P'_j}),
\end{aligned}$$

where step $a$ follows from the triangle inequality of Hamming distance; and step $b$ follows from the fact that $x^{1-\frac{1}{\rho_j}} \leq 1$ for $x \in [0,1]$ and $\frac{1}{\rho_j} \in [0,1]$ and $s < 1$. Similarly, we can verify that $\mathcal{E}_{Q_\sigma}\left(h'_{\sigma'}, h^*_{P'_j}\right) \leq \mathcal{E}_{P'_j}^{\frac{1}{\rho_j}}(h'_{\sigma'}, h^*_{P'_j})$ for $h'_{\sigma'} \in \tilde{\mathcal{H}}$, which means that

$$\mathcal{E}_{Q_\sigma}\left(h, h^*_{P'_j}\right) \leq \mathcal{E}_{P'_j}^{\frac{1}{\rho_j}}(h), \quad \forall h \in \tilde{\mathcal{H}}.$$

So $P'_j \to Q_\sigma$ satisfies the transfer-exponent with parameters $(\rho_j, 1)$ for all $\sigma \in \{-1, +1\}^{d-1}$ and $j \in [K]$.

**Verification of the fact $\mathcal{E}_Q\left(h_{P^*_j}\right) \leq \alpha_j$ for all $j \in [K]$ and $\sigma \in \{-1, +1\}^{d-1}$.**

- If $r_j = \varepsilon_j$, then for any $\sigma \in \{-1, +1\}^{d-1}$, we have $h^*_{P_{j,\sigma}} = h_\sigma$, then:

$$\mathcal{E}_{Q_\sigma}\left(h^*_{P_{j,\sigma}}\right) = \mathcal{E}_{Q_\sigma}(h_\sigma) = 0 \leq \alpha_j.$$

- If $r_j = \alpha_j$, then $h_{P'_j}^* = h_{\boldsymbol{\eta}_j}$, then for any $\boldsymbol{\sigma} \in \{-1, +1\}^{d-1}$, we have:

$$
\mathcal{E}_{Q_{\boldsymbol{\sigma}}}\left(h_{P'_j}^*\right) = \mathcal{E}_{Q_{\boldsymbol{\sigma}}}\left(h_{\boldsymbol{\eta}_j}\right) = \frac{H(\boldsymbol{\sigma}, \boldsymbol{\eta}_j)}{d-1} \cdot s \leq s
$$

$$
= c_1 \cdot s'_K \leq c_1 \cdot r_j = c_1 \cdot \alpha_j \overset{a}{\leq} \alpha_j,
$$

where step $a$ follows from the fact that $c_1 \in (0, 1)$.

So we have verified that $(P_{1,\boldsymbol{\sigma}}, \cdots, P_{K,\boldsymbol{\sigma}}, Q_{\boldsymbol{\sigma}}) \in \mathcal{P} \;\; \forall \boldsymbol{\sigma} \in \{-1, +1\}^{d-1}$. According to Lemma A.4, we know that there exists a subset $\Sigma \subseteq \{-1, +1\}^{d-1}$ with $M := |\Sigma| \geq 2^{\frac{d-1}{8}}$ such that $H(\boldsymbol{\sigma}_i, \boldsymbol{\sigma}_j) \geq \frac{d-1}{8}$ for any $\boldsymbol{\sigma}_i, \boldsymbol{\sigma}_j \in \Sigma$. It shows that for any $\boldsymbol{\sigma}, \boldsymbol{\sigma}' \in \Sigma$, we have:

$$
\mathcal{E}_{Q_{\boldsymbol{\sigma}}}(h_{\boldsymbol{\sigma}'}) = \frac{H(\boldsymbol{\sigma}, \boldsymbol{\sigma}')}{d-1} \cdot s \geq \frac{s}{8}. \tag{10}
$$

Let $I = \{j \in [K] : r_j = \varepsilon_j\}$, define $\Pi_{\boldsymbol{\sigma}} = P_{1,\boldsymbol{\sigma}}^{n_1} \times \cdots \times P_{K,\boldsymbol{\sigma}}^{n_K}$, then for any $\boldsymbol{\sigma}, \boldsymbol{\sigma}' \in \Sigma$, we have:

$$
D_{\mathrm{KL}}\left(\Pi_{\boldsymbol{\sigma}} \| \Pi_{\boldsymbol{\sigma}'}\right) = D_{\mathrm{KL}}\left(P_{1,\boldsymbol{\sigma}}^{n_1} \times \cdots \times P_{K,\boldsymbol{\sigma}}^{n_K} \| P_{1,\boldsymbol{\sigma}'}^{n_1} \times \cdots \times P_{K,\boldsymbol{\sigma}'}^{n_K}\right)
$$

$$
\overset{a}{=} \sum_{j=1}^{K} n_j \cdot D_{\mathrm{KL}}\left(P_{j,\boldsymbol{\sigma}} \| P_{j,\boldsymbol{\sigma}'}\right) \overset{b}{=} \sum_{j \in I} n_j \cdot D_{\mathrm{KL}}\left(P_{j,\boldsymbol{\sigma}} \| P_{j,\boldsymbol{\sigma}'}\right)
$$

$$
\overset{b}{=} \sum_{j \in I} n_j \cdot \underset{P_{j,X}}{\mathbb{E}}\left[D_{\mathrm{KL}}(P_{j,\boldsymbol{\sigma},Y|X} \| P_{j,\boldsymbol{\sigma}',Y|X})\right]
$$

$$
= \sum_{j \in I} n_j \left[\sum_{i=1}^{d-1} \frac{s^{\rho_j \beta_j}}{d-1} D_{\mathrm{KL}}(P_{j,\boldsymbol{\sigma},Y|X=x_i} \| P_{j,\boldsymbol{\sigma}',Y|X=x_i}) + \left(1 - s^{\rho_j \beta_j}\right) D_{\mathrm{KL}}(P_{j,\boldsymbol{\sigma},Y|X=x_0} \| P_{j,\boldsymbol{\sigma}',Y|X=x_0})\right]
$$

$$
\overset{c}{=} \sum_{j \in I} n_j \cdot \left[\sum_{i=1}^{d-1} \frac{s^{\rho_j \beta_j}}{d-1} \cdot D_{\mathrm{KL}}(P_{j,\boldsymbol{\sigma},Y|X=x_i} \| P_{j,\boldsymbol{\sigma}',Y|X=x_i})\right]
$$

$$
\overset{d}{\leq} \sum_{j \in I} n_j \cdot \left[\sum_{i=1}^{d-1} \frac{s^{\rho_j \beta_j}}{d-1} \cdot c_0 \cdot s^{2\rho_j(1-\beta_j)}\right]
$$

$$
= c_0 \cdot \sum_{j \in I} n_j \cdot s^{\rho_j \beta_j} \cdot s^{2\rho_j(1-\beta_j)}
$$

$$
= c_0 \cdot \sum_{j \in I} n_j \cdot s^{\rho_j(2-\beta_j)} = c_0 \cdot \sum_{j \in I} n_j \cdot c_1^{\rho_j(2-\beta_j)} \cdot (s'_K)^{\rho_j(2-\beta_j)}
$$

$$
\overset{e}{\leq} c_0 \cdot \sum_{j \in I} n_j \cdot c_1^{\rho_j(2-\beta_j)} \cdot d \overset{f}{\leq} K c_0 c_1 d,
$$

where step $a$ follows from the properties 1 and 2 in Lemma A.2; step $b$ follows from the property 3 in Lemma A.2; step $c$ follows from the fact that $D_{\mathrm{KL}}(P_{j,\boldsymbol{\sigma},Y|X=x_0} \| P_{j,\boldsymbol{\sigma}',Y|X=x_0}) = 0$; step $d$ follows from Lemma A.5; step $e$ follows from the fact that $s'_K \leq \varepsilon_j \; \forall j \in I$ and the definition of $\varepsilon_j$; and step $f$ follows from the fact that $\rho_j(2-\beta_j) \geq$ and $c_1 < 1$ implies that $c_1^{\rho_j(2-\beta_j)} \leq c_1$. We can choose $c_1 \in (0, 1)$ sufficiently small so that $K c_0 c_1 \leq \frac{\log 2}{72}$, which means that

$$
D_{\mathrm{KL}}\left(\Pi_{\boldsymbol{\sigma}} \| \Pi_{\boldsymbol{\sigma}'}\right) \leq \frac{d \log 2}{72} = \frac{1}{8} \cdot \frac{d \log 2}{9} \overset{a}{\leq} \frac{1}{8} \cdot \frac{1}{8} \cdot \frac{d-1}{8} \log 2 \overset{b}{\leq} \frac{1}{8} \log M,
$$

where step $a$ follows from the fact that $d \geq 9$ implies $d \leq \frac{9}{8} \cdot (d-1)$ and step $b$ follows from the fact that $M \geq 2^{\frac{d-1}{8}}$. Applying Lemma A.3 to the constructed family of distribution tuples $(P_{1,\boldsymbol{\sigma}}, \cdots, P_{K,\boldsymbol{\sigma}}, Q_{\boldsymbol{\sigma}})$ indexed by $\boldsymbol{\sigma} \in \Sigma$, we have know that for any improper learner $\hat{h}$:

$$
\sup_{\boldsymbol{\sigma} \in \Sigma} \underset{Z \sim \Pi_{\boldsymbol{\sigma}}}{\mathbb{P}}\left(\mathcal{E}_{Q_{\boldsymbol{\sigma}}}\left(\hat{h}(Z)\right) \geq \frac{s}{16}\right) \geq \frac{3 - 2\sqrt{2}}{8},
$$

which means that there exists some constant $c$ (depend on $c_0, K$) such that for any improper learner $\hat{h}$:

$$\sup_{(P_1,\ldots,P_K,Q)\in\mathcal{P}} \mathbb{P}_{S_{P_1}\sim P_1^{n_1},\ldots,S_{P_K}\sim P_K^{n_K}} \left[ \mathcal{E}_Q\left(\hat{h}(S_{P_1},\ldots,S_{P_K})\right) \geq c\cdot s_K \right] \geq \frac{3-2\sqrt{2}}{8}.$$

$\square$

# B. Proofs of the Upper Bounds

**Lemma B.1** (A Lemma from (Vapnik & Chervonenkis, 1974)). *Let* $A_n = \frac{d_{\mathcal{H}}}{n}\log\left(\frac{\max\{n,d_{\mathcal{H}}\}}{d_{\mathcal{H}}}\right) + \frac{1}{n}\log\left(\frac{1}{3\delta}\right)$, *where* $\mathcal{H}\subseteq\mathcal{Y}^{\mathcal{X}}$ *and* $d_{\mathcal{H}} = VCdim(\mathcal{H})$. *Let* $P$ *be a distribution over* $\mathcal{X}\times\mathcal{Y}$, *then with probability at least* $1-\delta$, $\forall\, h, h' \in \mathcal{H}$:

$$R_P(h) - R_P(h') \leq R_{\hat{P}}(h) - R_{\hat{P}}(h') + c\sqrt{\min\{P(h\neq h'),\hat{P}(h\neq h')\}\cdot A_n} + cA_n,$$

*and*

$$\frac{1}{2}P(h\neq h') - cA_n \leq \hat{P}(h\neq h') \leq 2P(h\neq h') + cA_n,$$

*for a universal constant* $c\in(0,\infty)$, *where* $\hat{P}$ *is the empirical distribution of* $n$ *i.i.d. samples from* $P$.

*Proof of Lemma 5.1.* Since $P$ satisfies the RBCC with parameters $(\beta, c_\beta)$, then there exists some $h_P^* \in \arg\min_{h\in\mathcal{H}} R_P(h)$ such that:

$$P_X(\hat{h}\neq h_P^*) \leq c_\beta \cdot \mathcal{E}_P^\beta(\hat{h}),$$

applying Lemma B.1 to $\hat{h}$ and $h_P^*$, we have: with probability at least $1-\delta$,

$$R_P\left(\hat{h}\right) - R_P(h_P^*) \leq R_{\hat{P}}\left(\hat{h}\right) - R_{\hat{P}}(h_P^*) + c\sqrt{P(\hat{h}\neq h_P^*)\cdot A_n} + cA_n$$

$$\leq R_{\hat{P}}\left(\hat{h}\right) - R_{\hat{P}}(h_P^*) + c\sqrt{c_\beta\cdot\mathcal{E}_P^\beta(\hat{h})\cdot A_n} + cA_n.$$

By the definition of $\hat{h}$, we have $R_{\hat{P}}\left(\hat{h}\right) - R_{\hat{P}}(h_P^*) \leq 0$, so with probability at least $1-\delta$:

$$\mathcal{E}_P\left(\hat{h}\right) = R_P\left(\hat{h}\right) - R_P(h_P^*) \leq c\sqrt{c_\beta\cdot\mathcal{E}_P^\beta(\hat{h})\cdot A_n} + cA_n.$$

Let $c' = \max\{c, c\sqrt{c_\beta}\}$, then with probability at least $1-\delta$:

$$\mathcal{E}_P\left(\hat{h}\right) \leq c'\sqrt{\mathcal{E}_P^\beta(\hat{h})\cdot A_n} + c'A_n. \tag{1}$$

If $A_n > 1$, then obviously the result holds, with $c_\rho = 1$. So without loss of generality we can assume that $A_n \leq 1$. We consider two situations:

- If $\sqrt{\mathcal{E}_P^\beta(\hat{h})\cdot A_n} \geq A_n$, then $\mathcal{E}_P\left(\hat{h}\right) \geq A_n^{\frac{1}{\beta}}$, combining Equation (1) we know that:

$$\mathcal{E}_P\left(\hat{h}\right) \leq c'\sqrt{\mathcal{E}_P^\beta(\hat{h})\cdot A_n} + c'A_n \leq 2c'\sqrt{\mathcal{E}_P^\beta(\hat{h})\cdot A_n},$$

which means that:

$$\mathcal{E}_P\left(\hat{h}\right) \leq (2c')^{\frac{2}{2-\beta}}\cdot A_n^{\frac{1}{2-\beta}}.$$

- If $\sqrt{\mathcal{E}_P^\beta(\hat{h})\cdot A_n} < A_n$, then $\mathcal{E}_P\left(\hat{h}\right) < A_n^{\frac{1}{\beta}}$. Since $\beta\in[0,1]$ and $A_n\in[0,1]$, we have $\beta\leq 2-\beta$ and then $A_n^{\frac{1}{\beta}} \leq A_n^{\frac{1}{2-\beta}}$. So we have:

$$\mathcal{E}_P\left(\hat{h}\right) \leq A_n^{\frac{1}{2-\beta}}.$$

In conclusion, there exists a constant $c'' = \max\left\{1, (2c')^{\frac{2}{2-\beta}}\right\} > 0$ such that with probability at least $1 - \delta$ we have:

$$\mathcal{E}_P\left(\hat{h}\right) \leq c'' \cdot A_n^{\frac{1}{2-\beta}}.$$

Since the transfer-exponent from $P$ to $Q$ is $\rho$ with parameter $C_\rho$, then

$$\mathcal{E}_Q(\hat{h}, h_P^*) \leq C_\rho \mathcal{E}_P^{\frac{1}{\rho}}(\hat{h}, h_P^*) \leq C_\rho \cdot (c'')^{\frac{1}{\rho}} \cdot A_n^{\frac{1}{\rho(2-\beta)}},$$

which means that there exists a constant $c_\rho := C_\rho \cdot (c'')^{\frac{1}{\rho}} > 0$ such that with probability at least $1 - \delta$ we have:

$$\mathcal{E}_Q(\hat{h}) \leq \mathcal{E}_Q(h_P^*) + c_\rho \cdot A_n^{\frac{1}{\rho(2-\beta)}}.$$

$\square$

*Proof of Theorem 5.2.* Theorem 5.2 is directly driven from the fact that Algorithm 1 knows $i^* = \arg\min_{i \in [K]} \left\{\alpha_i + c_{\rho,i} \cdot A_{n_i}^{\frac{1}{\rho_i(2-\beta_i)}}\right\}$ and Lemma 5.1. Here $C' = \max\{c_{\rho,1}, \ldots, c_{\rho,K}\}$ that depends on $\rho_i, \beta_i, C$. $\square$

*Proof of Theorem 5.3.* Let $C' = \max\{c_{\rho,1}, \ldots, c_{\rho,K}\}$, then by Lemma 5.1, we have that with probability at least $1 - \delta$,

$$\mathcal{E}_Q\left(\hat{h}\right) \leq \mathcal{E}_Q\left(h_{P_k}^*\right) + c_{\rho,k} \cdot A_{n_k}^{\frac{1}{\rho_k(2-\beta_k)}} \leq \mathcal{E}_Q\left(h_{P_k}^*\right) + C' \cdot A_{n_k}^{\frac{1}{\rho_k(2-\beta_k)}}.$$

Since We don't know the relative magnitude between

$$\mathcal{E}_Q\left(h_{P_1}^*\right) + C' \cdot A_{n_1}^{\frac{1}{\rho_1(2-\beta_1)}}, \ldots, \mathcal{E}_Q\left(h_{P_K}^*\right) + C' \cdot A_{n_K}^{\frac{1}{\rho_K(2-\beta_K)}},$$

there are some cases that

$$\mathcal{E}_Q\left(h_{P_k}^*\right) + C' \cdot A_{n_k}^{\frac{1}{\rho_k(2-\beta_k)}} = \max_{i \in [K]}\left\{\mathcal{E}_Q\left(h_{P_1}^*\right) + C' \cdot A_{n_1}^{\frac{1}{\rho_1(2-\beta_1)}}, \ldots, \mathcal{E}_Q\left(h_{P_K}^*\right) + C' \cdot A_{n_K}^{\frac{1}{\rho_K(2-\beta_K)}}\right\},$$

So we can only say that

$$\mathcal{E}_Q\left(\hat{h}\right) \leq \max_{i \in [K]}\left\{\mathcal{E}_Q\left(h_{P_1}^*\right) + C' \cdot A_{n_1}^{\frac{1}{\rho_1(2-\beta_1)}}, \ldots, \mathcal{E}_Q\left(h_{P_K}^*\right) + C' \cdot A_{n_K}^{\frac{1}{\rho_K(2-\beta_K)}}\right\}$$

rather than

$$\mathcal{E}_Q\left(\hat{h}\right) \leq \min_{i \in [K]}\left\{\mathcal{E}_Q\left(h_{P_1}^*\right) + C' \cdot A_{n_1}^{\frac{1}{\rho_1(2-\beta_1)}}, \ldots, \mathcal{E}_Q\left(h_{P_K}^*\right) + C' \cdot A_{n_K}^{\frac{1}{\rho_K(2-\beta_K)}}\right\}.$$

$\square$

*Proof of Theorem 5.4.* We first fix some $i \in [K]$ and apply Lemma 5.1 to $P_i$, we have that with probability at least $1 - \frac{\delta}{K}$:

$$\mathcal{E}_Q\left(h_{\hat{P}_i}\right) \leq \mathcal{E}_Q\left(h_{P_i}^*\right) + c_{\rho,i} \cdot A_{n_i,K}^{\frac{1}{\rho_i(2-\beta_i)}}.$$

Using the union bound for $i = 1, 2, \ldots, K$, we have that with probability at least $1 - \delta$:

$$\mathcal{E}_Q\left(h_{\hat{P}_i}\right) \leq \mathcal{E}_Q\left(h_{P_i}^*\right) + c_{\rho,i} \cdot A_{n_i,K}^{\frac{1}{\rho_i(2-\beta_i)}}, \quad \forall i \in [K].$$

Let $C' = \max\{c_{\rho,1}, \ldots, c_{\rho,K}\}$, then by the definition of $\hat{h} = \text{Rand}\left(h_{\hat{P}_1}, \ldots, h_{\hat{P}_K}\right)$, we know that condition on the above event (which has probability at least $1 - \delta$):

$$\mathcal{E}_Q\left(\hat{h}\right) = \frac{1}{K} \sum_{i=1}^{K} \mathcal{E}_Q\left(h_{\hat{P}_i}\right) \leq \frac{1}{K} \sum_{i=1}^{K} \left(\mathcal{E}_Q\left(h_{P_i}^*\right) + C' \cdot A_{n_i,K}^{\frac{1}{\rho_i(2-\beta_i)}}\right).$$

$\square$

*Proof of Lemma 5.7.* By Assumption 5.5, we choose $h_P^* = h_Q^* = h_R^* = h^*$. Since $P \to Q$ satisfies the transfer-exponent with parameters $(\rho_1, C_1)$, we have for any $h \in \mathcal{H}$:

$$\mathcal{E}_Q(h, h^*) = \mathcal{E}_Q(h, h_P^*) \leq C_1 \cdot \mathcal{E}_P^{\frac{1}{\rho_1}}(h, h_P^*) = C_1 \cdot \mathcal{E}_P^{\frac{1}{\rho_1}}(h, h^*).$$

Since $Q \to R$ satisfies the transfer-exponent with parameters $(\rho_2, C_2)$, then for any $h \in \mathcal{H}$:

$$\mathcal{E}_R(h, h^*) = \mathcal{E}_R(h, h_Q^*) \leq C_2 \cdot \mathcal{E}_Q^{\frac{1}{\rho_2}}(h, h_Q^*) = C_2 \cdot \mathcal{E}_Q^{\frac{1}{\rho_2}}(h, h^*).$$

Combining the above two equations, we can conclude that for any $h \in \mathcal{H}$:

$$\mathcal{E}_R(h, h_P^*) = \mathcal{E}_R(h, h^*) = \mathcal{E}_R(h, h_Q^*) \leq C_2 \cdot \mathcal{E}_Q^{\frac{1}{\rho_2}}(h, h_Q^*) = C_2 \cdot \mathcal{E}_Q^{\frac{1}{\rho_2}}(h, h^*)$$
$$= C_2 \cdot \mathcal{E}_Q^{\frac{1}{\rho_2}}(h, h_P^*) \leq C_2 \cdot (C_1)^{\frac{1}{\rho_2}} \, \mathcal{E}_P^{\frac{1}{\rho_1 \rho_2}}(h, h_P^*).$$

So $P \to R$ satisfies the transfer-exponent with parameters $\left(\rho_1 \rho_2, C_2 \cdot (C_1)^{\frac{1}{\rho_2}}\right)$. $\qquad\square$

*Proof of Theorem 5.9.* For any $j \in [K] \backslash \{1\}$, apply Lemma B.1 we know that on the event $E_j(S_{P_j})$ with $\mathbb{P}(E_j) \geq 1 - \frac{\delta}{K}$, let $h = h_{\hat{P}_j}$ and $h' = h_{P_1}^*$, we have

$$R_{P_j}\left(h_{\hat{P}_j}\right) - R_{P_j}\left(h_{P_1}^*\right) \leq R_{\hat{P}_j}\left(h_{\hat{P}_j}\right) - R_{\hat{P}_j}\left(h_{P_1}^*\right) + c\sqrt{\hat{P}_j(h_{\hat{P}_j} \neq h_{P_1}^*) \cdot A_{n_j, K}} + cA_{n_j, K}.$$

Since $\mathcal{E}_{P_j}\left(h_{P_1}^*\right) = 0$, $R_{P_j}\left(h_{\hat{P}_j}\right) - R_{P_j}\left(h_{P_1}^*\right) = R_{P_j}\left(h_{\hat{P}_j}\right) - R_{P_j}\left(h_{P_j}^*\right) \geq 0$, so we have:

$$R_{\hat{P}_j}\left(h_{P_1}^*\right) - R_{\hat{P}_j}\left(h_{\hat{P}_j}\right) \leq c\sqrt{\hat{P}_j(h_{\hat{P}_j} \neq h_{P_1}^*) \cdot A_{n_j, K}} + cA_{n_j, K},$$

which means that $h_{P_1}^* \in \mathcal{H}_j$ for $j \in [K] \backslash \{1\}$, so $h_{P_1}^* \in \cap_{j=2}^K \mathcal{H}_j$, then by Assumption 5.5 we know that $h_{P_1}^*, h^* \in \cap_{j=2}^K \mathcal{H}_j$, so $\cap_{j=2}^K \mathcal{H}_j$ is not empty and Algorithm 4 makes sense. Again, by Lemma B.1, on $E_j$ for $j \in [K] \backslash \{1\}$, we let $h = h_{\hat{P}_j}$ and $h' = h_{P_j}^*$ we have:

$$R_{P_j}\left(h_{\hat{P}_j}\right) - R_{P_j}\left(h_{P_j}^*\right) \leq R_{\hat{P}_j}\left(h_{\hat{P}_j}\right) - R_{\hat{P}_j}\left(h_{P_j}^*\right) + c\sqrt{P_j(h_{\hat{P}_j} \neq h_{P_j}^*) \cdot A_{n_j, K}} + cA_{n_j, K}.$$

Since $R_{\hat{P}_j}\left(h_{\hat{P}_j}\right) - R_{\hat{P}_j}\left(h_{P_j}^*\right) \leq 0$ by the definition of $h_{\hat{P}_j}$, we have:

$$\mathcal{E}_{P_j}\left(h_{\hat{P}_j}\right) = R_{P_j}\left(h_{\hat{P}_j}\right) - R_{P_j}\left(h_{P_j}^*\right) \leq c\sqrt{P_j(h_{\hat{P}_j} \neq h_{P_j}^*) \cdot A_{n_j, K}} + cA_{n_j, K}.$$

Since $P_j$ satisfies the RBCC with parameters $(\beta_j, c_j)$, the following holds on $E_j$:

$$\mathcal{E}_{P_j}\left(h_{\hat{P}_j}\right) \leq c\sqrt{c_j \cdot \mathcal{E}_{P_j}^{\beta_j}\left(h_{\hat{P}_j}\right) \cdot A_{n_j, K}} + cA_{n_j, K}.$$

Let $c_j' = \max\{c, c\sqrt{c_j}\}$, then by the same argument in the proof of Lemma 5.1, we know that on $E_j$:

$$\mathcal{E}_{P_j}\left(h_{\hat{P}_j}\right) \leq c_j'' \cdot A_{n_j, K}^{\frac{1}{2 - \beta_j}},$$

where $c_j'' = \max\left\{1, (2c_j')^{\frac{2}{2 - \beta_j}}\right\} > 0$. Following the analogous argument for $S_{P_1}$, it follows that for any set $\mathcal{G} \subseteq \mathcal{H}$ with $h_{P_1}^* \in \mathcal{G}$, on a event $E_1$ where $\mathbb{P}(E_1) \geq 1 - \frac{\delta}{K}$, the ERM $h_{\hat{P}_1}' = \arg\min_{h \in \mathcal{G}} R_{\hat{P}_1}(h)$ satisfies:

$$\mathcal{E}_{P_1}\left(h_{\hat{P}_1}'\right) \leq c_1'' \cdot A_{n_1, K}^{\frac{1}{2 - \beta_1}}.$$

We set $\mathcal{G} = \cap_{j=2}^{K} \mathcal{H}_j$, then on the event $\cap_{j=2}^{K} E_j$, we have that $h^*_{P_1}, h^* \in \mathcal{G}$, so on the event $\cap_{j=2}^{K} E_j \cap E_1$ we have that the output of Algorithm 4 $\hat{h} = h'_{\hat{P}_1}$ satisfies:

$$\mathcal{E}_{P_1}\left(\hat{h}\right) \le c''_1 \cdot A_{n_1,K}^{\frac{1}{2-\beta_1}}.$$

If $\bar{\rho}_j = \infty$ or $A_{n_j,K} > 1$, the bounds trivially hold, so we consider the case that $\bar{\rho}_j < \infty$ and $A_{n_j,K} \le 1$. On event $E_j$, let $h = \hat{h}$ and $h' = h_{\hat{P}_j}$, we have:

$$R_{P_j}\left(\hat{h}\right) - R_{P_j}\left(h_{\hat{P}_j}\right) \le R_{\hat{P}_j}\left(\hat{h}\right) - R_{\hat{P}_j}\left(h_{\hat{P}_j}\right) + c\sqrt{\hat{P}_j(\hat{h} \ne h_{\hat{P}_j}) \cdot A_{n_j,K}} + cA_{n_j,K}$$
$$\overset{a}{\le} 2c\sqrt{\hat{P}_j(\hat{h} \ne h_{\hat{P}_j}) \cdot A_{n_j,K}} + 2cA_{n_j,K},$$

where step $a$ follows from the fact that $\hat{h} \in \mathcal{H}_j$, which means that $R_{\hat{P}_j}\left(\hat{h}\right) - R_{\hat{P}_j}\left(h_{\hat{P}_j}\right) \le c\sqrt{\hat{P}_j(\hat{h} \ne h_{\hat{P}_j}) \cdot A_{n_j,K}} + cA_{n_j,K}$. We have:

$$\sqrt{\hat{P}_j(\hat{h} \ne h_{\hat{P}_j}) \cdot A_{n_j,K}} + A_{n_j,K} \le \sqrt{\left(2P_j(\hat{h} \ne h_{\hat{P}_j}) + cA_{n_j,K}\right) \cdot A_{n_j,K}} + A_{n_j,K}$$
$$\overset{a}{\le} \sqrt{2P_j(\hat{h} \ne h_{\hat{P}_j}) \cdot A_{n_j,K}} + \sqrt{c}A_{n_j,K} + A_{n_j,K}$$
$$\overset{b}{\le} \underbrace{\sqrt{2P_j(\hat{h} \ne h^*_{P_j}) \cdot A_{n_j,K}}}_{(I)} + \underbrace{\sqrt{2P_j(h^*_{P_j} \ne h_{\hat{P}_j}) \cdot A_{n_j,K}}}_{(II)} + \left(\sqrt{c} + 1\right) A_{n_j,K}$$

where step $a$ follows from the fact that $\sqrt{a+b} \le \sqrt{a} + \sqrt{b}$ when $a, b \ge 0$; step $b$ come from $\mathbb{1}\left[\hat{h}(x) \ne h_{\hat{P}_j}(x)\right] \le \mathbb{1}\left[\hat{h}(x) \ne h^*_{P_j}(x)\right] + \mathbb{1}\left[h^*_{P_j}(x) \ne h_{\hat{P}_j}(x)\right]$ for any $x$. Since $P_j$ satisfies the RBCC with parameters $(\beta_j, c_j)$, we have $(I) \le \sqrt{2c_j \mathcal{E}_{P_j}^{\beta_j}(\hat{h}) \cdot A_{n_j,K}}$ and:

$$(II) \le \sqrt{2c_j \mathcal{E}_{P_j}^{\beta_j}(h_{\hat{P}_j}) \cdot A_{n_j,K}} \overset{a}{\le} \sqrt{2c_j c''^{\beta_j}_j A_{n_j,K}^{\frac{\beta_j}{2-\beta_j}} \cdot A_{n_j,K}} = \sqrt{2c_j c''^{\beta_j}_j} A_{n_j,K}^{\frac{1}{2-\beta_j}},$$

where step $a$ is from the same argument in Lemma 5.1. Combing the above, we have that:

$$R_{P_j}\left(\hat{h}\right) - R_{P_j}\left(h_{\hat{P}_j}\right) \le 2c\sqrt{2c_j \mathcal{E}_{P_j}^{\beta_j}(\hat{h}) \cdot A_{n_j,K}} + 2c\sqrt{2c_j c''^{\beta_j}_j} A_{n_j,K}^{\frac{1}{2-\beta_j}} + 2c\left(\sqrt{c} + 1\right) A_{n_j,K}$$
$$\overset{a}{\le} 2c\sqrt{2c_j \mathcal{E}_{P_j}^{\beta_j}(\hat{h}) \cdot A_{n_j,K}} + 2c\left(\sqrt{2c_j c''^{\beta_j}_j} + \sqrt{c} + 1\right) A_{n_j,K}^{\frac{1}{2-\beta_j}},$$

where step $a$ is from the fact that $x^{\frac{1}{2-\beta_j}} \ge x$ when $x \in [0, 1]$ and $\beta_j \in [0, 1]$. Then we have that on $E_j$:

$$\mathcal{E}_{P_j}\left(\hat{h}\right) = R_{P_j}\left(\hat{h}\right) - R_{P_j}\left(h_{\hat{P}_j}\right) + \mathcal{E}_{P_j}\left(h_{\hat{P}_j}\right)$$
$$\overset{a}{\le} 2c\sqrt{2c_j \mathcal{E}_{P_j}^{\beta_j}(\hat{h}) \cdot A_{n_j,K}} + 2c\left(\sqrt{2c_j c''^{\beta_j}_j} + \sqrt{c} + 1\right) A_{n_j,K}^{\frac{1}{2-\beta_j}} + c''_j A_{n_j,K}^{\frac{1}{2-\beta_j}} \tag{2}$$
$$\overset{b}{\le} c_j^{(3)} \sqrt{\mathcal{E}_{P_j}^{\beta_j}(\hat{h}) \cdot A_{n_j,K}} + c_j^{(3)} A_{n_j,K}^{\frac{1}{2-\beta_j}},$$

where step $a$ is from the same argument in Lemma 5.1 and step $b$ is obtained by defining $c_j^{(3)} = \max\left\{2c\left(\sqrt{2c_j c''^{\beta_j}_j} + \sqrt{c} + 1\right) + c''_j, 2c\sqrt{2c_j}\right\} > 0$. Based on Equation (2), we consider two case:

- If $\sqrt{\mathcal{E}_{P_j}^{\beta_j}(\hat{h}) \cdot A_{n_j,K}} \ge A_{n_j,K}^{\frac{1}{2-\beta_j}}$, then we have $\mathcal{E}_{P_j}\left(\hat{h}\right) \le 2c_j^{(3)} \sqrt{\mathcal{E}_{P_j}^{\beta_j}(\hat{h}) \cdot A_{n_j,K}}$, which means that:

$$\mathcal{E}_{P_j}\left(\hat{h}\right) \le \left(2c_j^{(3)}\right)^{\frac{2}{2-\beta_j}} \cdot A_{n_j,K}^{\frac{1}{2-\beta_j}}.$$

- If $\sqrt{\mathcal{E}_{P_j}^{\beta_j}(\hat{h}) \cdot A_{n_j,K}} \leq A_{n_j,K}^{\frac{1}{2-\beta_j}}$, then we have $\mathcal{E}_{P_j}(\hat{h}) \leq A_{n_j,K}^{\frac{1}{2-\beta_j}}$.

In conclusion, if we set $c_j^{(4)} = \max\left\{1, \left(2c_j^{(3)}\right)^{\frac{2}{2-\beta_j}}\right\}$, then we have:

$$\mathcal{E}_{P_j}\left(\hat{h}\right) \leq c_j^{(4)} \cdot A_{n_j,K}^{\frac{1}{2-\beta_j}}.$$

Since the transfer-exponent from $P_j$ to $P_1$ is $\bar{\rho}_j$ with parameter $C_{\bar{\rho}_j}$, then:

$$\mathcal{E}_{P_1}\left(\hat{h}\right) \leq c_j^{(4)} \cdot C_{\bar{\rho}_j} \cdot A_{n_j,K}^{\frac{1}{(2-\beta_j)\bar{\rho}_j}}.$$

Define $C_1 = \max\left\{c_2^{(4)}C_{\bar{\rho}_2}, \ldots, c_K^{(4)}C_{\bar{\rho}_K}, c_1''\right\}$, we then have that on the event $\cap_{j=1}^K E_j$:

$$\mathcal{E}_{P_1}\left(\hat{h}\right) \leq C_1 \cdot \min_{j\in[K]}\left\{A_{n_j,K}^{\frac{1}{(2-\beta_j)\bar{\rho}_j}}\right\}.$$

Since the transfer-exponent from $P_1$ to $Q$ is $\rho_1$ with parameter $C_{\rho_1}$, then by Assumption 5.5, with probability at least $1 - \delta$:

$$\mathcal{E}_Q(\hat{h}) \leq C' \cdot \min_{j\in[K]}\left\{A_{n_j,K}^{\frac{1}{(2-\beta_j)\bar{\rho}_j\rho_1}}\right\},$$

where $C' = C_1 \cdot C_{\rho_1}$. $\qquad\square$

Before proving Theorem 5.12, we need to define an auxiliary distribution tuple class and an auxiliary algorithm.

**Definition B.2** ($\mathcal{Q}$-Class Related to the $\mathcal{P}$ Class). Let $\rho_1, \ldots, \rho_K \in (0, \infty], \beta_1, \ldots, \beta_K, \beta \in [0, 1], C > 0$. Then there exists $i^* = \arg\min_{i\in[K]} \rho_i$, we say $\mathcal{Q}(\mathcal{H}; \bar{\rho}_1, \ldots, \bar{\rho}_K; \beta_1, \ldots, \beta_K; \beta; C)$ is the $\mathcal{Q}$-class related to $\mathcal{P}(\mathcal{H}; \rho_1, \ldots, \rho_K; \beta_1, \ldots, \beta_K; \beta; 0, \ldots, 0; C)$ if $\bar{\rho}_j = \frac{\rho_j}{\rho_{i^*}}$ for all $j \in [K]$ and $\mathcal{Q}(\mathcal{H}; \bar{\rho}_1, \ldots, \bar{\rho}_K; \beta_1, \ldots, \beta_K; \beta; C)$ contains all distribution tuples $(P_1, \ldots, P_K, Q)$ that satisfy the following conditions:

- For all $i \in [K]$, $P_i$ satisfies the RBCC with parameters $(\beta_i, c_i)$ with $c_i \leq C$. $Q$ satisfies the RBCC with parameters $(\beta, c)$ with $c \leq C$.

- For all $i \in [K]\backslash\{i^*\}$, the transfer exponent from $P_i$ to $P_{i^*}$ with respect to $\mathcal{H}$ is $\bar{\rho}_i$, with the corresponding constant $C_{\bar{\rho}_i} \leq C$. The transfer exponent from $P_{i^*}$ to $Q$ with respect to $\mathcal{H}$ is $\rho_{i^*}$, with the corresponding constant $C_{\rho_{i^*}} \leq C$.

**Lemma B.3.** *If $\mathcal{Q}$ is the $\mathcal{Q}$-class related to $\mathcal{P}$, then we have $\mathcal{Q} \subseteq \mathcal{P}$.*

*Proof of Lemma B.3.* If $(P_1, \ldots, P_K, Q) \in \mathcal{Q}$, then for all $j \neq i^*$, the transfer exponent from $P_j$ to $P_{i^*}$ is $\bar{\rho}_j$ and the transfer exponent from $P_{i^*}$ to $Q$ is $\rho_{i^*}$, then by Lemma 5.7, the transfer exponent from $P_j$ to $Q$ is $\bar{\rho}_j \cdot \rho_{i^*} = \rho_j$. So we can conclude that $(P_1, \ldots, P_K, Q) \in \mathcal{P}$. $\qquad\square$

The auxiliary algorithm is a generalization of Algorithm 4, which can achieve nearly optimal error rate given the information of $i^*$.

**Lemma B.4** (Lower Bound for the $\mathcal{Q}$-class). *Let $\mathcal{H}$ be a hypothesis class with VC dimension $VCdim(\mathcal{H}) = d \geq 9$. Fix any $\rho_1, \ldots, \rho_K \in [1, \infty], \beta_1, \ldots, \beta_K, \beta \in [0, 1]$. Let $\mathcal{Q}$ be the $\mathcal{Q}$-class related to $\mathcal{P}(\mathcal{H}; \rho_1, \ldots, \rho_K; \beta_1, \ldots, \beta_K; \beta; 0, \ldots, 0; 1)$. Let $\hat{h} = \hat{h}(S_{P_1}, \ldots, S_{P_K})$ denote any (possibly improper) classifier with access to $K$ independent labeled samples $S_{P_1} \sim P_1^{n_1}, \ldots, S_{P_K} \sim P_K^{n_K}$. Suppose that one of $\{n_1, \ldots, n_K\}$ is sufficiently large so that*

$$\varepsilon(n_1, \ldots, n_K) := \min\left\{\left(\frac{d}{n_1}\right)^{\frac{1}{(2-\beta_1)\rho_1}}, \ldots, \left(\frac{d}{n_K}\right)^{\frac{1}{(2-\beta_K)\rho_K}}\right\} \leq 1.$$

*Then, for any $\hat{h}$, there exists $(P_1, \ldots, P_K, Q) \in \mathcal{Q}$ and a universal constant $c > 0$ such that*

$$\mathbb{P}_{S_{P_1} \sim P_1^{n_1}, \ldots, S_{P_K} \sim P_K^{n_K}}\left(\mathcal{E}_Q\left(\hat{h}\right) > c \cdot \varepsilon(n_1, \ldots, n_K)\right) \geq \frac{3 - 2\sqrt{2}}{8}.$$

---

**Algorithm 6** A Generalization of Algorithm 4

---

**Input:** $\mathcal{H}$, $\{S_{P_i}\}_{i=1}^{K}$, $i^* = \arg\min_{i \in [K]} \rho_i$, and constant $c$ in Lemma B.1.

**for** $j \in [K] \backslash \{i^*\}$ **do**

Set $\mathcal{H}_j = \left\{ h \in \mathcal{H} : R_{\hat{P}_j}(h) - R_{\hat{P}_j}\left(h_{\hat{P}_j}\right) \leq c \sqrt{\hat{P}_j\left(h \neq h_{\hat{P}_j}\right) A_{n_j, K}} + c A_{n_j, K} \right\}.$

**end for**

Choose $\hat{h} \in \arg\min_{h \in \cap_{j \in [K] \backslash \{i^*\}} \mathcal{H}_j} R_{\hat{P}_{i^*}}(h).$

**Output:** $\hat{h}.$

---

*Remark* B.5. Note that for any distribution tuple $(P_1, \ldots, P_K, Q)$ that satisfies Assumption 5.5, we can find a group of parameters $\rho_1, \ldots, \rho_K \in [1, \infty], \beta_1, \ldots, \beta_K, \beta \in [0, 1], C > 0$ so that $(P_1, \ldots, P_K, Q)$ belongs to the $\mathcal{Q}$-class related to $\mathcal{P}(\mathcal{H}; \rho_1, \ldots, \rho_K; \beta_1, \ldots, \beta_K; \beta; 0, \ldots, 0; C)$. So Lemma B.4 applies to all distribution tuples with proper parameters and is very general.

*Proof of Lemma B.4.* Let $\mathcal{P} = \mathcal{P}(\mathcal{H}; \rho_1, \ldots, \rho_K; \beta_1, \ldots, \beta_K; \beta; 0, \ldots, 0; 1)$ and $\mathcal{Q}$ be the $\mathcal{Q}$-class related to $\mathcal{P}$. The crucial step of the proof is the observation that when $\alpha_1 = \cdots = \alpha_K = 0$, the sequence of distribution tuples $(P_{1, \boldsymbol{\sigma}}, \cdots, P_{K, \boldsymbol{\sigma}}, Q_{\boldsymbol{\sigma}}) \in \mathcal{P}$ $\forall \boldsymbol{\sigma} \in \{-1, +1\}^{d-1}$ constructed in the proof of Theorem 4.2 also belongs to $\mathcal{Q}$, and then we can use the same argument in the proof of Theorem 4.2 to get the lower bound. Then it remains to prove that $(P_{1, \boldsymbol{\sigma}}, \cdots, P_{K, \boldsymbol{\sigma}}, Q_{\boldsymbol{\sigma}}) \in \mathcal{Q}$ for any $\boldsymbol{\sigma} \in \{-1, +1\}^{d-1}$.

Fix any $\boldsymbol{\sigma} \in \{-1, +1\}^{d-1}$, from the proof of Theorem 4.2, it is clear that the RBCC condition holds and that $P_{i^*, \boldsymbol{\sigma}} \to Q_{\boldsymbol{\sigma}}$ satisfies the transfer-exponent with parameters $(\rho_{i^*}, 1)$. It remains to prove that $P_{j, \boldsymbol{\sigma}} \to P_{i^*, \boldsymbol{\sigma}}$ satisfies the transfer-exponent with parameters $\left(\frac{\rho_j}{\rho_{i^*}}, 1\right)$. From the proof of Theorem 4.2, for any $j \in [K] \backslash \{i^*\}$, we have

$$
\begin{aligned}
\mathcal{E}_{P_{i^*}}^{\frac{\rho_j}{\rho_{i^*}}}(h_{\boldsymbol{\sigma}'}, h_{P_{j, \boldsymbol{\sigma}}}^*) &= \left( \frac{H(\boldsymbol{\sigma}, \boldsymbol{\sigma}')}{d-1} \cdot s^{\rho_{i^*}} \right)^{\frac{\rho_j}{\rho_{i^*}}} \\
&= \left( \frac{H(\boldsymbol{\sigma}, \boldsymbol{\sigma}')}{d-1} \right)^{\frac{\rho_j}{\rho_{i^*}} - 1} \cdot \frac{H(\boldsymbol{\sigma}, \boldsymbol{\sigma}')}{d-1} \cdot s^{\rho_j} \\
&\overset{a}{\leq} \frac{H(\boldsymbol{\sigma}, \boldsymbol{\sigma}')}{d-1} \cdot s^{\rho_j} = \mathcal{E}_{P_{i^*}}(h_{\boldsymbol{\sigma}'}, h_{P_{j, \boldsymbol{\sigma}}}^*),
\end{aligned}
$$

where step $a$ follows from Definition B.2, where $\frac{\rho_j}{\rho_{i^*}} \geq 1$, and the fact that $x^{k-1} \leq 1$ for $x \in [0, 1]$ and $k \geq 1$. Similarly can verify that $\mathcal{E}_{P_{i^*}}^{\frac{\rho_j}{\rho_{i^*}}}(h'_{\boldsymbol{\sigma}'}, h_{P_{j, \boldsymbol{\sigma}}}^*) \leq \mathcal{E}_{P_{i^*}}(h'_{\boldsymbol{\sigma}'}, h_{P_{j, \boldsymbol{\sigma}}}^*)$, so $P_{j, \boldsymbol{\sigma}} \to P_{i^*, \boldsymbol{\sigma}}$ satisfies the transfer-exponent with parameters $\left(\frac{\rho_j}{\rho_{i^*}}, 1\right)$. $\qquad \square$

**Lemma B.6** (Upper Bound for Algorithm 6). *Let $\mathcal{H}$ be a hypothesis class with VC dimension $VCdim(\mathcal{H}) = d \geq 9$. Fix any $\rho_1, \ldots, \rho_K \in [1, \infty], \beta_1, \ldots, \beta_K, \beta \in [0, 1], C > 0$. Let $\mathcal{Q}$ be the $\mathcal{Q}$-class related to $\mathcal{P}(\mathcal{H}; \rho_1, \ldots, \rho_K; \beta_1, \ldots, \beta_K; \beta; 0, \ldots, 0; C)$ and let $(P_1, \ldots, P_K, Q) \in \mathcal{Q}$. If Assumption 5.5 holds and let $\hat{h}$ be the output of Algorithm 6, then there exists a constant $C' > 0$ (depending on $\rho_i, \beta_i, C$) such that with probability at least $1 - \delta$,*

$$
\mathcal{E}_Q\left(\hat{h}\right) \leq C' \cdot \min_{j \in [K]} \left\{ A_{n_j, K}^{\frac{1}{(2 - \beta_j)\rho_j}} \right\}.
$$

*Proof of Lemma B.6.* The proof can be derived from the fact that $\bar{\rho}_j = \frac{\rho_j}{\rho_{i^*}}$ and a similar argument as in the proof of Theorem 5.9, with the role of 1 and $i^*$ switched. $\qquad \square$

Now we are ready to prove Theorem 5.12.

*Proof of Theorem 5.12.* The lower bound is directly from Lemma B.4. Now we prove the upper bound. According to the empirical RCS condition, $\cap_{i=1}^{K} \hat{\mathcal{H}}_i \neq \emptyset$, so there exists a $\hat{h} \in \mathcal{H}$ so that $\hat{h} \in \arg\min_{h \in \mathcal{H}} R_{\hat{P}_i}(h)$ for all $i \in [K]$, which means that $\arg\min_{h \in \mathcal{H}} \frac{1}{K} \sum_{k=1}^{K} R_{\hat{P}_k}(h) \neq \emptyset$, so Algorithm 5 makes sense. Let $\hat{h}_{\text{ERM}} \in \arg\min_{h \in \mathcal{H}} \frac{1}{K} \sum_{k=1}^{K} R_{\hat{P}_k}(h)$ be the output of Algorithm 5. On the one hand, in the proof of Theorem 5.9, we know that any ERM learner of $\hat{P}_j$ belongs to $\mathcal{H}_j$ defined in Algorithm 6, so $\hat{h}_{\text{ERM}} \in \cap_{j \in [K] \setminus \{i^*\}} \mathcal{H}_j$. On the other hand, by the definition of $\hat{h}_{\text{ERM}}$, we have $\hat{h}_{\text{ERM}} \in \arg\min_{h \in \mathcal{H}} R_{\hat{P}_{i^*}}(h)$. So we conclude that $\hat{h}_{\text{ERM}} \in \arg\min_{h \in \cap_{j \in [K] \setminus \{i^*\}} \mathcal{H}_j} R_{\hat{P}_{i^*}}(h)$, which means that $\hat{h}_{\text{ERM}}$ can be regarded as the output of Algorithm 6 without knowing $i^*$. So according to Lemma B.6, the upper bound of Algorithm 5 is proved. $\qquad\square$

## C. Discussions about the Proof Techniques

Since our theory uses the transfer-exponent defined in (Hanneke & Kpotufe, 2019) to describe the ability to transfer information from one domain to another, in this section, we show the hardness of the analyzing OOD generalization and point out that the techniques in (Hanneke & Kpotufe, 2019; Hanneke et al., 2023) can not be easily applied to the OOD generalization setting.

Firstly, the settings considered are different. (Hanneke & Kpotufe, 2019; Hanneke et al., 2023) consider transfer learning where we have access to data from the source domain $P$ and target domain $Q$, the aim is to provide bounds for the target excess risk. However, this paper considers OOD generalization, where we have **no** access to the data from the target domain $Q$ and can make use of data from **multiple** source domains $P_1, \ldots, P_K$. Due to different kind of input information of the algorithm, the analysis of transfer learning can not be easily extended to OOD generalization.

### C.1. The Lower Bound

Let $P_1$ be the source domain and $Q$ be the target domain in (Hanneke & Kpotufe, 2019; Hanneke et al., 2023). We define $\varepsilon_1 = \left(\frac{d}{n_1}\right)^{\frac{1}{(2-\beta_1)\rho_1}}$ and $\alpha_1 = \mathcal{E}_Q(h_{P_1}^*)$, which are exactly the same as defined in Theorem 4.2. Similarly, we define $\varepsilon_Q = \left(\frac{d}{n_Q}\right)^{\frac{1}{(2-\beta_Q)}}$ and $\alpha_Q = \mathcal{E}_Q(h_Q^*) = 0$.

THE DIFFICULTY OF GENERALIZATION TO OOD SETTING

The aim of (Hanneke & Kpotufe, 2019; Hanneke et al., 2023) is to prove that the lower bound is of order $\Omega(\min\{\varepsilon_1 + \alpha_1, \varepsilon_Q\})$. To prove this, they separately construct two classes of distribution tuples to prove the lower bound is of order $\Omega(\min\{\varepsilon_1, \varepsilon_Q\})$ and $\Omega(\min\{\alpha_1, \varepsilon_Q\})$. Combining the results of the two constructions, they can prove that the lower bound is of order $\Omega(\max\{\min\{\varepsilon_1, \varepsilon_Q\}, \min\{\alpha_1, \varepsilon_Q\}\})$. Then, by the fact that

$$\frac{1}{2}\min\{\varepsilon_1 + \alpha_1, \varepsilon_Q\} \leq \max\{\min\{\varepsilon_1, \varepsilon_Q\}, \min\{\alpha_1, \varepsilon_Q\}\} \leq \min\{\varepsilon_1 + \alpha_1, \varepsilon_Q\}, \tag{3}$$

they can conclude that the lower bound is of order $\Omega(\min\{\varepsilon_1 + \alpha_1, \varepsilon_Q\})$. However, such a proof strategy does not work in our setting. Note that we need to prove a lower bound of order $\Omega\left(\min_{i \in [K]}\{\varepsilon_i + \alpha_i\}\right)$, the multiple source domains make it hard get a relationship like Equation (3). For example , the most direct consideration is to prove that $\min_{i \in [K]}\{\varepsilon_i + \alpha_i\}$ and $\max\left\{\min_{i \in [K]}\{\varepsilon_i\}, \min_{i \in [K]}\{\alpha_i\}\right\}$ are of the same order. However, since $\min_{i \in [K]}\{\varepsilon_i + \alpha_i\}$ and $\min_{i \in [K]}\{\max\{\varepsilon_i, \alpha_i\}\}$ are of the same order, we need to prove that $\min_{i \in [K]}\{\max\{\varepsilon_i, \alpha_i\}\}$ and $\max\left\{\min_{i \in [K]}\{\varepsilon_i\}, \min_{i \in [K]}\{\alpha_i\}\right\}$ are of the same order. The above argument reduces to a problem of proving $\min_j \max_i A_{ij}$ and $\max_i \min_j A_{ij}$ are of the same order for a matrix $A \in [0, 1]^{2 \times K}$. Unfortunately, such a claim does not hold generally since for

$$A = \begin{pmatrix} 1 & 0 & 1 \\ 0 & 1 & 0 \end{pmatrix},$$

we have $\min_j \max_i A_{ij} = 1$ and $\max_i \min_j A_{ij} = 0$. So we can not find a constant $c > 0$ where $\max_i \min_j A_{ij} \geq c \cdot \min_j \max_i A_{ij}$, which means that $\min_j \max_i A_{ij}$ and $\max_i \min_j A_{ij}$ are of the same order generally.

OUR SOLUTION

Since directly generalizing the technique in (Hanneke & Kpotufe, 2019; Hanneke et al., 2023) to OOD generalization setting is difficult, we choose to construct the distribution tuples according to the characteristics of OOD generalization problem. In this paper, we construct $P_j$ according to the value of $\varepsilon_j$ and $\alpha_j$, rather than separately construct tuples for $\varepsilon_j$ and $\alpha_j$ respectively (the method in (Hanneke & Kpotufe, 2019; Hanneke et al., 2023)). So our constructed tuples are quite different from those in (Hanneke & Kpotufe, 2019; Hanneke et al., 2023). Moreover, our new construction method makes the tuples hard to satisfy the BCC condition (Definition 3.1), which is required in (Hanneke & Kpotufe, 2019; Hanneke et al., 2023). Fortunately, we find that our construction satisfies a relaxed version of BCC (Definition 3.2), which is enough for the proof of the upper bounds.

The above shows that the proof of Theorem 4.2 is non-trivial. Furthermore, the class of distribuion tuples constucted in Theorem 4.2 is called the $\mathcal{P}$-class (Definition 4.1), while in Theorem 5.12, to prove the ERM is nearly optimal, we construct the $\mathcal{Q}$-class (Definition B.2), which is again much more different from those in (Hanneke & Kpotufe, 2019; Hanneke et al., 2023) compared to the $\mathcal{P}$-class. **In conclusion, our construction differs from that in (Hanneke & Kpotufe, 2019; Hanneke et al., 2023) a lot and our lower bounds are novel**.

## C.2. The Upper Bound

The algorithms in (Hanneke & Kpotufe, 2019) are different from ours. Most importantly, they can use the target data, while our algorithm has only access to the source data. Due to the lower bound (Theorem 4.2), the target excess risk is of order $\Omega\left(\min_{i\in[K]}\left\{\left(\frac{d}{n_i}\right)^{\frac{1}{(2-\beta_i)\rho_i}}\right\}\right)$. As noted in Section 1.2, Lemma 5.1 and Theorem 4.2 show that the knowledge of $i^*$ can help us design a nearly optimal algorithm. However, the lack of distributional parameters makes it difficult to find an algorithm that reaches a rate of $\tilde{O}\left(\min_{i\in[K]}\left\{\left(\frac{d}{n_i}\right)^{\frac{1}{(2-\beta_i)\rho_i}}\right\}\right)$ (see explorations in Section 5). Surprisingly, we find that a simple ERM algorithm can reach a nearly optimal rate under some mild conditions. **In conclusion, our ERM algorithm is quite different from the algorithms in (Hanneke & Kpotufe, 2019) and our results are novel**.

