# OpenReview forum: "Generalization Bounds for Out-of-distribution Generalization"
_ICML.cc/2026/Conference — ICML 2026 regular_

### Official Review · Reviewer_UN2z · 2026-03-09

**Soundness:** 3
**Presentation:** 3
**Significance:** 3
**Originality:** 3
**Overall Recommendation:** 5
**Confidence:** 3

**Summary:**

The submission investigates theoretical guarantees for generalization to out-of-distribution (OOD) data by studying the statistical limits of OOD generalization. It further explains why many sophisticated OOD algorithms fail to outperform empirical risk minimization (ERM), analyzing this phenomenon through the lens of statistical learning theory. Specifically, the submission derives both lower and upper bounds on the target excess risk when models are trained on multiple source distributions and evaluated on an unseen target distribution. The analysis leverages tools such as VC dimension, the Bernstein class condition, and transfer exponents. The theoretical results show that minimizing the average empirical risk across source domains can achieve nearly optimal error rates under RCS assumptions, providing a theoretical explanation for empirical observations, such as those reported in DomainBed, where ERM performs competitively with specialized OOD algorithms.

**Compliance With Llm Reviewing Policy:**

Affirmed.

**Final Justification:**

The paper is technically sound and offers a useful way to characterize the upper and lower bounds of OOD generalization. Its main weaknesses lie in the unclear motivation and the strong assumptions.

The rebuttal addressed my primary concerns regarding both the motivation and the assumptions. However, the absence of a more principled approach to relaxing these assumptions prevents me from assigning a score higher than 5 (Accept).

**Key Questions For Authors:**

C1 and C2 are my main concerns. Please carefully answer these two questions. If my comments have any misunderstandings or mistakes, please feel free to correct me. Thanks.

**Limitations:**

yes

**Strengths And Weaknesses:**

**Strengths:**

++ The submission provides both lower and upper bounds on OOD generalization, offering a more complete statistical characterization of the problem.

++ The theoretical results offer a compelling explanation for why ERM often performs competitively with specialized OOD algorithms in benchmarks such as DomainBed.

++ The submission presents clear definitions and proofs using established tools from statistical learning theory, including VC dimension analysis and minimax arguments.

++ By connecting transfer-exponent analysis with OOD generalization, the work introduces a new theoretical angle that may inspire further research on the limits of domain generalization.



**Weaknesses:**

-- The submission does not fully justify why analyzing minimax bounds for OOD generalization from the aspect of statistical learning theory. See C1.

-- The theoretical results rely on assumptions that may limit applicability. See C2.

**Detailed Comments:**

C1: While the submission frames two fundamental questions regarding the upper and lower bounds of target excess risk in OOD generalization, the motivation for prioritizing this theoretical perspective could be strengthened. The OOD literature already contains several theoretical frameworks (e.g., invariant risk minimization, distributionally robust optimization, and spurious feature analysis), and the submission does not clearly argue why the statistical learning theory view adopted here provides deeper insights than these existing directions. In particular, the connection between the proposed bounds and practical OOD algorithm design is not direct enough.



C2: As a conditional theoretical result, the use of the RCS assumption is acceptable. However, it is much less convincing when used to support the broader practical claim that ERM is nearly optimal for OOD generalization. Assumption 4.5 is quite strong, as it requires the existence of a single hypothesis h^* that simultaneously minimizes the risk on all source domains as well as the target domain. This appears to overlook one primary challenge of OOD generalization: different domains may favor different predictors due to spurious correlations, conditional label shifts, or other domain-specific structures. A similar concern applies to the empirical counterpart of this condition: satisfying an empirical condition over the training domains does not constitute a population-level guarantee. In particular, achieving zero training loss across source domains does not, by itself, imply that the same classifier is optimal for the unseen target domain. Although the submission presents this as a special theoretical regime rather than a universal model of OOD, this still limits the practical applicability of the takeaways.



C3: I believe it may be worthwhile to explore whether the upper-bound regime could be interpreted through a hierarchical Bayesian perspective, which might soften the exact one-optimum-hypothesis assumptions and make the framework more practically relevant. Such a perspective could more naturally model partial sharing across domains than the current formulation.


C4: Although the submission states that the related work discussion is deferred to Appendix A, I believe it would be more appropriate to include at least a concise related work section in the main body of the submission. This would make it easier for readers to assess the positioning of the work during the main reading flow. If the submission is accepted, I encourage the authors to move at least the most relevant discussion of related work into the main content.

---

> ### Author Rebuttal · Authors · 2026-03-25
>
> Thank you very much for the thoughtful review and for the constructive comments. We also appreciate that you identified C1 and C2 as the main concerns, and we answer them carefully below.
>
> ## **Answer to C1**
>
> We agree that this motivation should be stated more clearly. We will make the following motivation more explicit in the revision.
>
> A key advantage of the statistical learning theory (SLT) / PAC perspective is that it focuses directly on the statistical limits of the OOD generalization task itself. In particular, it allows us to ask:
> - **how hard the OOD task is in principle**, through minimax lower bounds;
> - and **what rates are achievable**, through upper bounds for concrete algorithms.
>
> The SLT framework provides **distribution-agnostic** results by providing **worst-case** measure of the performance, where the worst-case is taken over the distributions. An advantage of this viewpoint is that we can get an elementary evaluation of a task $\mathcal{H}\subseteq\mathcal{Y}^\mathcal{X}$ without requiring explicit knowledge of the source-target relationship. The upper bounds show that, over the considered distribution class and under the stated conditions, a concrete algorithm can attain the corresponding rate.
>
> It's hard to tell which framework is better, and they all have their limitations. The problem with a distribution-agnostic framework is that it is conservative due to the worst-case measure. And this might be one of the reasons that the optimal algorithms in the SLT framework are not used in practice (**answering the question "In particular, the connection between the proposed bounds and practical OOD algorithm design is not direct enough"**). The reason is that in practice, our aim is to get good performance on some given datasets, but not all possible distributions. However, it might be useful in some situations that require strong security, where we need to evaluate the performance on all possible distributions. The problem with other frameworks that assume a specific structure of the source-target distributions is that the performance of the proposed methods is not guaranteed when we meet distributions that violate the assumption. So, **the SLT framework provides a different perspective compared to existing frameworks, and we believe that it's worth studying**.
>
> As for the new insight from the SLT framework, the near optimality of ERM in our regime also helps explain why ERM is competitive in DomainBed. More precisely, the near optimality in the worst-case sense means that no algorithm can **uniformly** achieve a better worst-case rate than ERM over the considered class, which is consistent with the observation in DomainBed.
>
> ## **Answer to C2**
>
> We agree with the reviewer that Assumption 4.5 is a strong assumption from the practical perspective, due to spurious correlations, conditional label shifts, or other domain-specific structures. In the discussion section, lines 403-420, the right column, we have discussed the relationship between the RCS assumption and other commonly used assumptions in the theoretical study of distributional shifts. The RCS assumption is neither strictly stronger nor strictly weaker than the well-studied covariate shift and concept shift assumptions, so it can be regarded as **a complementary perspective to these assumptions**. We appreciate that the reviewer thinks "as a conditional theoretical result, the use of the RCS assumption is acceptable".
>
> Regarding the fact that empirical RCS does not imply population-level RCS. Thank you for pointing it out. In fact, we have noticed that, and this is the reason Theorem 4.12 needs both empirical RCS and RCS, but not only empirical RCS. We are glad to make it clearer in our revision.
>
> In total, we agree that the RCS condition, **a complement of commonly used covariate shift and concept shift assumptions**, is strong in practice, and some current wording in the paper is broader than intended. In the revision, we will make the scope explicit and revise these statements to clarify that our conclusion should be read as: **ERM is nearly optimal in the RCS / empirical-RCS regime**, rather than as a universal practical claim about OOD generalization in deep learning. The main limitation of this paper is the RCS assumption in the upper bound. Nevertheless, we believe that, as a theoretical paper, our results still provide meaningful insights from the statistical learning theory perspective.
>
> ## **Answer to C3 and C4**
> We thank the reviewer for these helpful suggestions. We agree that this is a promising direction and that some related work should be included in the main body. We will mention the former as an important future direction (and try to study it in the future) and move the related work discussion to the main text in the revision.

---

> > ### Author Rebuttal · Reviewer_UN2z · 2026-04-01
> >
> > Thank you to the authors for the careful and thoughtful rebuttal. My concerns have been adequately addressed. However, given the limitations of the RCS assumption and the absence of a more principled and elegant way to relax it, 5 (Accept) is the highest score I can assign. Again, thank you for presenting this work, and I genuinely enjoyed reading it. Best of luck.

---

> > > ### Author Response · Authors · 2026-04-01
> > >
> > > Dear Reviewer,
> > >
> > > Thank you very much for your positive assessment and for your thoughtful comments. We are sincerely grateful for your time, effort, and constructive suggestions throughout the review process.

---

### Official Review · Reviewer_85wZ · 2026-03-10

**Soundness:** 3
**Presentation:** 3
**Significance:** 3
**Originality:** 3
**Overall Recommendation:** 5
**Confidence:** 3

**Summary:**

The authors provide lower and upper bounds on the excess risk of a target distribution $Q$, given $K$ source distributions $P_i$. They make use of the Bernstein class condition and transfer exponents to characterize the relationship between $P_i$ and $Q$ and to derive these bounds. This differs from prior work that typically relies on distribution divergence. The bounds are then used to establish a no-free-lunch theorem for OOD generalization and to explain why no OOD algorithm consistently outperforms ERM when combined with a powerful hypothesis class, such as neural networks, as evidenced in DomainBed.

**Compliance With Llm Reviewing Policy:**

Affirmed.

**Final Justification:**

The paper provides, to my knowledge, an alternative view and mathematical toolkit for analyzing bounds on OOD generalization. The authors’ response clarified the points I previously raised, which stemmed from my misunderstanding of the framework. Therefore, I'm keeping my positive recommendation.

**Key Questions For Authors:**

You state that ERM is a nearly optimal OOD algorithm under mild assumptions (e.g., Assumption 4.5), which ideally can be achieved with neural networks on many domains/datasets. Does the no-free-lunch theorem for OOD still apply under these assumptions? More broadly, and at the risk of sounding naive, would you say that your results remove the need to design or apply OOD deep learning algorithms if we believe that Assumption 4.5 holds?

**Limitations:**

Yes

**Strengths And Weaknesses:**

This is an interesting paper.

**Soundness**: This work is a little outside my expertise, but the theorems seem sound.

**Presentation**: The presentation is okay. It is quite heavy with notation that seems somewhat obtuse, but it all seems necessary, and I couldn't think of any way of simplifying it further. I'm sure part of this is due to my unfamiliarity with the material. I will say that the Preliminary section is quite good.

**Significance and Originality**: I think the utilization of the Bernstein class condition (BCC) and transfer exponents is useful and original, as it deviates from other approaches that utilize optimization [1] or divergence [2] to explain OOD generalization. It is particularly significant that both a lower and an upper bound are defined and used to explain ERM’s empirical dominance. I do wonder whether there are limitations to this approach. Please address the key question section.

[1] Invariant Risk Minimization, 2020, Martin Arjovsky et. al.

[2] A theory of learning from different domains, 2010, Shai Ben-David et. al.

---

> ### Author Rebuttal · Authors · 2026-03-25
>
> Thank you very much for the positive assessment and the insightful questions. About the heavy notations, we will try our best to make them more readable in our revision.
>
> Your key questions have two parts. The first part concerns whether the no-free-lunch theorem for OOD still holds under Assumption 4.5, and the second part wonders whether our results remove the need to design or apply OOD deep learning algorithms if we believe that Assumption 4.5 holds. We will answer these two questions separately.
>
> ## **Question 1 - Does the no-free-lunch theorem still apply under Assumption 4.5?**
>
> The no-free-lunch theorem **still holds** under Assumption 4.5. In fact, **the no-free-lunch theorem is a result of the lower bound**. Under Assumption 4.5, our lower bound (please refer to the lower bound in Theorem 4.12) becomes $\Omega \left( \min_ i \left( \frac{d}{n_ i} \right)^{\frac{1}{(2-\beta_ i) \rho_ i}} \right)$, so the no-free-lunch theorem holds as a result of the lower bound. More precisely, as stated in Remark 3.4, in this situation, we can choose distributions such that $\rho_ 1 = \cdots = \rho_ K = \infty$, then the excess risk is lower bounded away from zero, which means that the no-free-lunch theorem holds. In fact, in lines 166-169, the right column, we have claimed that the no-free-lunch theorem holds even when there exists a hypothesis $h^* \in \mathcal{H}$ that simultaneously minimizes both the source and target risks, which is exactly equivalent to Assumption 4.5. We are sorry for the unclear statement, and we are glad to clarify this point in our revision.
>
>
> ## **Question 2 - Do our results remove the need to design or apply OOD deep learning algorithms?**
>
> We would say that our results **do not** remove the need to design OOD deep learning algorithms. Our results show that ERM is nearly optimal under some mild assumptions **in the minimax / worst-case sense**. More precisely, the lower bound characterizes the best possible worst-case target excess-risk rate achievable by any algorithm, while Theorem 4.12 shows that ERM attains this rate up to logarithmic factors.
>
> Therefore, the near optimality of ERM means that, without additional information about the relationship between the source and target distributions, one cannot hope to design an algorithm that **uniformly** achieves a better worst-case rate than ERM. This means that for an algorithm $A$ that does not achieve nearly optimal performance, it can not **uniformly** achieve a better performance than ERM, i.e., we can find a distribution tuple $(P_ 1, \dots, P_ K, Q) \in \mathcal{P}$ such that ERM has better performance on $(P_ 1, \dots, P_ K, Q)$ compared with $A$. However, this **does not rule out** the possibility that other OOD algorithms may outperform ERM on particular datasets or under **additional structure**. In fact, this also helps explain why, as discussed in Remark 4.15, carefully designed OOD methods may outperform ERM on some datasets, yet underperform ERM on other datasets.
>
> In this sense, our results suggest that if we have no further information about the target domain, then there is very limited room to improve upon ERM in a uniform worst-case sense. However, when **additional structure** on the source-target relationship is available, it may still be possible to design algorithms that outperform ERM **in those more specialized scenarios**.

---

> > ### Author Rebuttal · Reviewer_85wZ · 2026-04-03
> >
> > Thank you. I appreciate your incisive response. I will maintain my positive assessment.

---

> > > ### Author Response · Authors · 2026-04-04
> > >
> > > Dear Reviewer,
> > >
> > > We greatly appreciate your positive feedback and thoughtful comments. Thank you for the time and care you invested in the review process, and for the constructive suggestions you provided.

---

### Official Review · Reviewer_BQue · 2026-03-11

**Soundness:** 3
**Presentation:** 3
**Significance:** 2
**Originality:** 2
**Overall Recommendation:** 4
**Confidence:** 4

**Summary:**

The paper first shows that, under the RBCC assumption, the excess risk of a target-domain classifier can be controlled by the excess risk on the source domain through the transfer exponent. Building on this relationship, the authors establish a probabilistic lower bound and argue that no single algorithm can guarantee generalization performance uniformly across all possible target domains.

The paper then derives upper bounds on the excess risk in the single-source setting and presents a single-source ERM algorithm that achieves the optimal rate. Furthermore, under sufficiently informative conditions, the authors extend the analysis to the multi-source setting, providing guarantees for arbitrary ERM and randomized ERM procedures, as well as a constrained ERM algorithm that achieves a nearly optimal rate.

Finally, under the Empirical RCS Assumption, the paper derives an error bound for the classifier obtained by minimizing the average empirical loss over the source domains, and shows that its upper bound also achieves a nearly optimal convergence rate.

**Compliance With Llm Reviewing Policy:**

Affirmed.

**Final Justification:**

The discussion on the significance of the theoretical contributions, including the characterization beyond the final ERM form, the empirical RCS condition, the proof details of Theorem 3.2, and the justification for the lower-bound statement, has clarified several issues raised in my earlier review. Considering these clarifications and the proposed revision, I am inclined to reconsider my score.

**Key Questions For Authors:**

Please see Strengths And Weaknesses.

**Limitations:**

Please see Strengths And Weaknesses.

**Strengths And Weaknesses:**

The paper is clearly organized and logically structured. The arguments are presented in a step-by-step manner, and the necessary assumptions and definitions are introduced with appropriate explanations.

The overall significance is moderate. The final results achieve a nearly optimal rate, and the corresponding algorithm appears to follow rather directly from the transfer exponent framework together with the Empirical RCS Assumption.

The technical development is generally credible. However, the proof of Theorem 3.2 should be completed with full details. In addition, in Remark 4.10, the claim that “we can adapt the lower bound for Pas a lower bound for Q” requires a separate and explicit justification.
Moreover, since the main guarantees are described as “nearly optimal,” the paper should further clarify why this gap is meaningful.

---

> ### Author Rebuttal · Authors · 2026-03-26
>
> Thank you very much for the careful reading and for the constructive comments. We address your concerns below.
>
> ## **About the significance**
>
> We understand the reviewer’s concern about the significance of the ERM algorithm. However, we would like to emphasize that the main novelty is not only the final algorithmic form, but the full characterization:
> - a minimax lower bound for multi-source OOD generalization (Theorem 3.2)
> - a separate lower bound for the $\mathcal{Q}$-class via Lemma C.4
> - and an upper bound showing that simple ERM matches the statistical limit up to logarithmic factors under the RCS / empirical-RCS regime
>
> Moreover, we believe that the proposal of the empirical RCS condition is nontrivial. After carefully analyzing the property of the OOD generalization task, we propose the empirical RCS condition to handle the problem that we do not know $i^* = \arg\min_ i \rho_ i$, a new issue arose in the multi-source situation.
>
> We also discuss in Appendix D that both the lower-bound tuple construction and the upper-bound analysis are nontrivial extensions beyond prior transfer-exponent arguments. We will make this novelty clearer in the revision.
>
>
> ## **About the full proof of Theorem 3.2**
>
> In the main paper, we only display the proof sketch of Theorem 3.2, and the full proof details are deferred to Appendix B due to page limitations. In line 180, the right column, we have pointed out that the full proof can be found in Appendix B. We are sorry for confusing you, and we will clarify this point in our revision.
>
> ## **About the claim “we can adapt the lower bound for P as a lower bound for Q” in Remark 4.10**
>
> Thank you for pointing this out. The explicit justification of this statement is in the proof of the lower bound in Theorem 4.12. More precisely, the lower bound for $\mathcal{Q}$ is in the proof of Lemma C.4. We are sorry for confusing you about this point. Due to page limitations, we cannot put the proof in Remark 4.10, but we are glad to clearly point out in Remark 4.10 that the full proof of this statement can be found in the proof of Lemma C.4, so that the justification is easier to locate.
>
>
> ## **Why is the “nearly optimal” gap meaningful?**
>
> Our lower bound is of order $\Omega \left( \min_ i \left( \frac{d}{n_ i} \right)^{\frac{1}{(2-\beta_ i) \rho_ i}} \right)$ and our upper bound is of order $\tilde{O} \left( \min_ i \left( \frac{d}{n_ i} \right)^{\frac{1}{(2-\beta_ i) \rho_ i}} \right)$. Here, the $\tilde{O}(\cdot)$ notation hides some logarithmic factors. There are several reasons that we believe this gap is meaningful.
> - The main parts of our upper and lower bounds, i.e., $\left( \frac{d}{n_ i} \right)^{\frac{1}{(2-\beta_ i) \rho_ i}}$, are polynomial in $n_ i$ and $d$, while the gap is logarithmic, which is negligible compared to the polynomial term. More precisely, the polynomial term $m^\alpha$ ($\alpha>0$) grows much faster than the $\log m$ term.
> - Our paper takes the first step to study OOD generalization in the Probably Approximately Correct (PAC) learning framework. Our nearly optimal rate is a starting point for the exploration of the optimal rate, and we believe this work still provides meaningful insights for characterizing OOD generalization from the statistical learning perspective.

---

> > ### Author Rebuttal · Reviewer_BQue · 2026-04-06
> >
> > Thank you very much for the authors’ insightful response on the theoretical aspects, which clarified some of the issues raised in the previous discussion. I will take this into consideration when revising my evaluation.

---

> > > ### Author Response · Authors · 2026-04-06
> > >
> > > Dear Reviewer,
> > >
> > > Thank you for your reply. We are glad that our response helped clarify some of the issues raised in the previous discussion. We would also like to thank you again for your time, effort, and insightful comments during the review process.

---

### Official Review · Reviewer_AuY5 · 2026-03-11

**Soundness:** 3
**Presentation:** 2
**Significance:** 4
**Originality:** 3
**Overall Recommendation:** 5
**Confidence:** 3

**Summary:**

This paper studies generalization bounds for binary classification problem on a target distribution $Q$ given access to data drawn from $K$ source distributions $P_1,\dots,P_K$ where each sample size $n_i,\forall i\in[K]$ is given a-priori. The main contributions consist of a high probability lower bound on the excess risk evaluated on the target distribution, target distribution excess risk upper bounds for the Oracle and ERM, AD-ERM, randomized ERM and constrained ERM algorithms.

**Compliance With Llm Reviewing Policy:**

Affirmed.

**Final Justification:**

I remain largely positive about the contribution of this work. My concerns are mostly about the presentation of the paper, which the authors adequately addressed in the rebuttal.

**Key Questions For Authors:**

1. In the proof of the lower bound where you applied Lemma B.3, am I correct to say that the semi-metric space you chose have $H:\big<\sigma,\sigma' \big>\mapsto \mathcal{E}\_{Q\_\sigma}(h\_{\sigma'})$  and the underlying set $\mathcal{F}$ is the same as the target of the hypothesis class $\mathcal{H}$?


2. In your paper you assume that the sample sizes $n_i$ are given a-priori but this is probably not true in practical settings. Will the analysis differ much if I allow $n_i$ to be randomly distributed by some categorical distribution?

**Limitations:**

Yes. Authors have adequately discussed limitations.

**Strengths And Weaknesses:**

**Soundness**: The submission is technically sound, all results are supported by proper proofs given in the appendix.

**Presentation**: I appreciate the author rigor. However, for layman's benefit, I think the intuition behind the technical definitions should be supplied alongside them. For example, the intuition (as I understand) behind $\beta$ in BCC can be described as the control of how fast the region of disagreement between a good classifier and an optimal one shrinks as excess risk decreases (please clarify since I am not too sure myself).

**Originality**: I think relating OOD generalization bounds to BCC and the transfer exponents is novel, especially the proof of the lower bound. I enjoy the idea of constructing a tuples subset in $\mathcal{P}$ indexed by Rademacher sequences then apply the result in [Tsykakov'09].

---

> ### Author Rebuttal · Authors · 2026-03-25
>
> Thank you very much for the positive assessment and the helpful suggestions on the presentation. Your intuition about the $\beta$ in BCC is correct. We agree that the intuition behind the technical definitions should be supplied alongside them, and we will add the intuitions for other definitions in our revision.
>
> ## **For Question 1**
>
> Your viewpoint is **intuitively correct, but there are some differences in the details**.
>
> - In fact, the underlying semi-metric space is not $\mathcal{H}$ directly. Rather, we first restrict our attention to the shattered finite set $\mathcal{S} = \\{ x_ 0, \dots, x_ {d-1} \\} \subseteq X$, and then set $\mathcal{F}$ as the projection of $\mathcal{H}$ onto $\mathcal{S}$, i.e., we set $\mathcal{F} = \tilde{\mathcal{H}} = \\{ (h(x_ 0), \dots, h(x_ {d-1})): h \in \mathcal{H} \\}$. This is because the constructed distribution class only supports on $\mathcal{S}$ and we do not care the performance of $h \in \mathcal{H}$ outside $\mathcal{S}$.
> - The semi-metric $H$ we choose is the Hamming distance on $\tilde{\mathcal{H}}$, i.e., $H(h_ \sigma, h_ \sigma') = \sum_ {i=1}^{d-1} 1[h_ \sigma(x_ i) \ne h_ \sigma' (x_ i)]$, which is defined in line 712. We write $H(\sigma, \sigma')$ instead of $H(h_ \sigma, h_ \sigma')$ for simplicity. We then use equation (10) in line 892 to convert the lower bound for the Hamming distance in Lemma B.3 into the lower bound for the target excess risk.
>
> We are sorry for confusing you, and we will clarify this point in our revision.
>
> ## **For Question 2**
>
> Thank you very much for this very good question. **When sample sizes $n_i$ are random, the main proof ideas should still apply after conditioning on the realized counts $N_i$**. Our current upper bounds are of the form $\mathbb{P}_ {S} [\mathcal{E}(A(S)) \le \epsilon(n_ 1, \dots, n_ K)] \ge 1-\delta$, and the lower bounds are of the form $\mathbb{P}_ {S} [\mathcal{E}(A(S)) > \epsilon(n_ 1, \dots, n_ K)] \ge c$. Take the upper bounds as an example (analogous conditioning arguments should apply to the lower-bound proof as well), we can first condition on $(N_ 1, \dots, N_ K)$ and apply our main proof ideas, and then take expectation with respect to $(N_ 1, \dots, N_ K)$ to get $\mathbb{P}_ {N_ 1, \dots, N_ K, S} [\mathcal{E}(A(S)) \le \epsilon(N_ 1, \dots, N_ K)] \ge 1-\delta$. In this case, the error bound $\epsilon(N_ 1, \dots, N_ K)$ is random and depends on the random training set sizes $N_ 1, \dots, N_ K$.
>
> We would also like to kindly note that treating per-domain sample sizes as fixed and known is a standard simplification in theoretical learning analyses. Our goal here is to isolate the statistical difficulty of the multi-source OOD problem. Extending the results to random sample sizes is a natural direction, but would require additional arguments and is beyond the current scope.

---

> > ### Author Rebuttal · Reviewer_AuY5 · 2026-04-03
> >
> > I thank the authors for their response, which resolves all my remaining concerns. As such, I will maintain my positive rating.

---

> > > ### Author Response · Authors · 2026-04-04
> > >
> > > Dear Reviewer,
> > >
> > > Thank you very much for your positive evaluation and for your insightful comments. We sincerely appreciate the time and effort you devoted to reviewing our paper, as well as your constructive suggestions throughout the process.

---

### Decision · Program_Chairs · 2026-04-30

**Decision:**

Accept (regular)

**Comment:**

This paper presents a theoretical analysis for the problem of generalization for out-of-distribution data.
The paper provides a lower and upper bounds on the target excess risk depending  on different scenarios and hypothesis, using possibly multiple source domains.
Additionally, they provide a result showing that the average empirical risk across source domains can achieve nearly optimal error rates under some RCS assumption.


Reviewers have appreciated the soundess of the work and its originality. Some weaknesses were raised requiring more justifications and explications or questioning the significance of the result giving some assumptions.
After rebuttal all reviewers have indicated that their concerns have been fully resolved.
3 reviewers gave a score of accept and one weak accept.

This is overall a solid contribution for ICML.
I propose then acceptance.